# Early life adversity decreases pre-adolescent fear expression by accelerating amygdala PV cell development

Gabriela Manzano Nieves[1], Marilyn Bravo[1], Saba Baskoylu[1], Kevin G Bath[2]*

[1]Department of Neuroscience, Brown University, Providence, United States; [2]Department of Cognitive, Linguistic, and Psychological Sciences, Brown University, Providence, United States

**Abstract** Early life adversity (ELA) is associated with increased risk for stress-related disorders later in life. The link between ELA and risk for psychopathology is well established but the developmental mechanisms remain unclear. Using a mouse model of resource insecurity, limited bedding (LB), we tested the effects of LB on the development of fear learning and neuronal structures involved in emotional regulation, the medial prefrontal cortex (mPFC) and basolateral amygdala (BLA). LB delayed the ability of peri-weanling (21 days old) mice to express, but not form, an auditory conditioned fear memory. LB accelerated the developmental emergence of parvalbumin (PV)-positive cells in the BLA and increased anatomical connections between PL and BLA. Fear expression in LB mice was rescued through optogenetic inactivation of PV-positive cells in the BLA. The current results provide a model of transiently blunted emotional reactivity in early development, with latent fear-associated memories emerging later in adolescence.

*For correspondence:
Kevin_Bath@Brown.edu

**Competing interests:** The authors declare that no competing interests exist.

## Introduction

Early life adversity (ELA) increases the lifetime risk for multiple forms of psychopathology, including anxiety disorders and major depressive disorder (*Agid et al., 1999*; *Draijer and Langeland, 1999*; *Heim and Nemeroff, 2001*; *Koenen and Widom, 2009*; *Widom, 1999*). ELA is associated with higher rates of negative outcomes than similar events experienced in adulthood (*Felitti et al., 1998*; *Heim and Nemeroff, 2001*; *McCauley et al., 1997*; *Mullen et al., 1996*; *Salmon and Bryant, 2002*). The increased lifetime risk for psychopathology is proposed to be the result of alterations in the developmental trajectories of brain centers regulating emotional learning and emotional expression. In agreement with this prediction, ELA has been shown to drive early engagement of the basolateral amygdala (BLA), a key node supporting emotional processing, threat assessment, and fear learning (*Bath et al., 2016*; *Moriceau et al., 2009*).

The neural circuit(s) supporting fear and aversive learning in adult rodents have been well characterized. Numerous studies have shown that the prelimbic (PL) subregion of the medial prefrontal cortex (mPFC) projects to the BLA (*Do-Monte et al., 2015b*; *Vertes, 2004*) and is necessary for fear retrieval (*Burgos-Robles et al., 2009*; *Courtin et al., 2014*; *Sierra-Mercado et al., 2011*), whereas the infralimbic (IL) subregion of mPFC supports extinction learning (*Adhikari et al., 2015*; *Do-Monte et al., 2015a*; *Sierra-Mercado et al., 2011*). Over early development, PL projections into amygdala begin to emerge around postnatal day (PND) 7, increasing through adolescence (*Cunningham et al., 2002*), and being pruned back during early adulthood (*Bouwmeester et al., 2002*; *Cressman et al., 2010*). The relatively late integration of the PL into the threat learning circuit may explain recent reports suggesting that the infant PL, unlike adult PL, is not involved in the

expression of sustained fear responses (*Chan et al., 2011*). Investigating the impact of ELA on the development of the fear circuit will allow us to simultaneously identify potential mechanisms promoting developmental changes in learned aversive behaviors and possible underlying changes that may contribute to ELA-associated effects on anxiety, depressive, and threat learning behavior.

Recently, a rodent model of resource insecurity, limited bedding and nesting (LB), was developed to simulate aspects of resource insecurity and altered parental care in human populations (*Bolton et al., 2019*; *Rice et al., 2008*). LB has been shown to induce significant distress in the dam, alter patterns of maternal behavior (*Bolton et al., 2019*; *Gallo et al., 2019*; *McLoyd, 1998*; *Rice et al., 2008*), and delay developmental processes such as sexual maturation and physical growth (*Manzano Nieves et al., 2019*; *Yam et al., 2017*). In addition, LB impacts behavioral outcomes with effects that persist into adulthood, increasing depressive-like behaviors (*Goodwill et al., 2019*), altering behavioral response to stress (*Cohen et al., 2013*; *Manzano-Nieves et al., 2018*), and affecting hippocampal-dependent learning (*Bath et al., 2016*; *Bath et al., 2017*; *Manzano-Nieves et al., 2018*; *Wang et al., 2011*). However, the effects of LB on the development of neuronal populations and regions supporting fear learning and expression are less well understood. In this study, we tested the impact of LB on the development of subregions of mPFC, BLA, projections from mPFC to BLA, as well as their effect on the acquisition and expression of fear learning across early development.

Here, we demonstrate that LB delayed physical development in mice and altered developmental expression of fear learning. Specifically, LB resulted in decreased body weight and brain weight from infancy into pre-adolescence. Changes in physical development were accompanied by an increase in parvalbumin (PV)-positive cell density in the BLA of peri-weanling mice and sex-specific changes in mPFC to BLA anatomical connectivity during peri-adolescence. During the period of elevated PV-positive cell density, mice transiently failed to exhibit threat-associated freezing behavior, with expression of this learned fear re-emerging during the peri-adolescent and adolescent period. Furthermore, the diminished fear expression during the peri-weaning period could be rescued through optogenetic inactivation of PV-positive cells in the BLA. Based upon these results, an ELA-associated premature increase in PV-positive cells in BLA as associated with a transient decrease in the expression, but not acquisition of fear learning.

## Results

### Early life adversity decreases physical growth

In humans, ELA is associated with changes in expected weight (*Hult et al., 2010*; *Maniam et al., 2014*; *Rondó et al., 2003*; *Wainstock et al., 2013*), with low weight in infancy predicting poorer cognitive outcomes (*Corbett and Drewett, 2004*; *Strathearn et al., 2001*). Thus, weight may serve as a biomarker for altered neurodevelopment and later risk for pathology. To model ELA in the form of resource insecurity, dams and pups were placed in conditions of low bedding and nesting materials (LB) from PND 4–11 (*Bolton et al., 2019*; *McLoyd, 1998*; *Rice et al., 2008*). To assess the impact of LB on somatic and brain development, the body weight and brain weight of control (Ctrl) and LB reared mice was measured at select ages across early development (PND 16, 21, 28, and 35). We found that LB reared female mice weighed significantly less than control mice at PND 16, 21, and 28, with differences diminishing by PND 35 (*Figure 1B* top), while LB reared males weighed significantly less than control male mice at PND 21 and 28 (*Figure 1B* bottom). Examination of brain weight in females (*Figure 1C* top) revealed a significant decrease in brain weight at PND 21, 28 and 35. LB males had decreased brain weight when compared to control males at PND 16, 21, and 28, with differences resolved by PND 35 (*Figure 1C* bottom). We predicted that ELA may shift resources away from somatic development to spare brain development. To test this prediction, the brain-to-body weight ratio was calculated across developmental time points (*Figure 1D*). LB reared female mice had a higher brain-to-body weight ratio than female control mice at PND 16, 21 and 28, while LB males had an increased brain-to-body ratio at PND 21 and 28. Thus LB effects on brain and total body weight were not proportional, with a greater impact on the development of the body at select developmental time points.

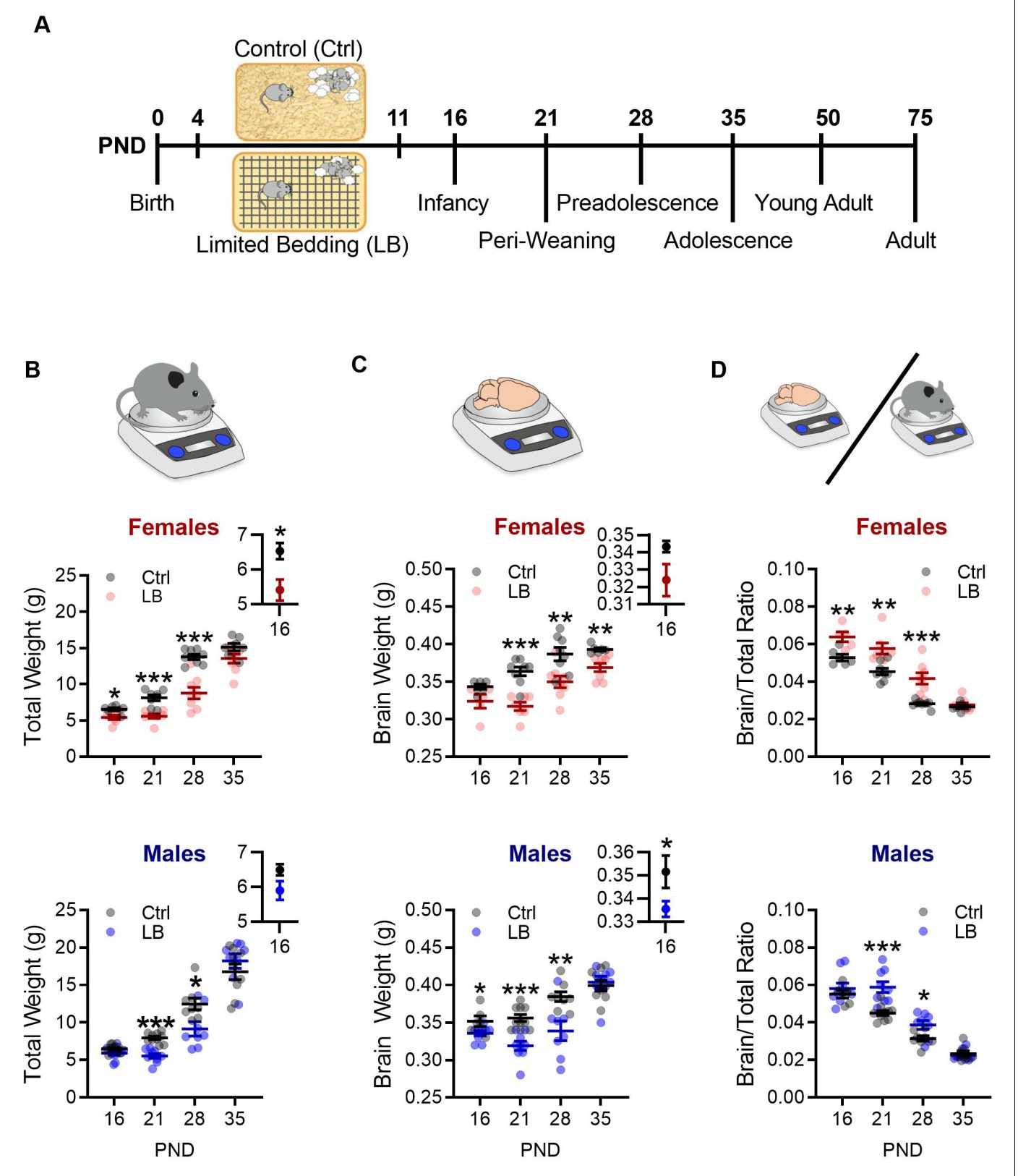

**Figure 1.** Limited bedding (LB) resources early in life, altered somatic and brain development. (A) Timeline of resource restriction manipulation (PND 4–11) and description of developmental time points tested. Somatic effects of LB rearing on Males and Females. (B) Graphs depicting LB changes in total body weight for females (top) and males (bottom). Insets show a close up of mean and SEM for PND 16 data. When compared to ctrl females, LB

*Figure 1 continued on next page*

Figure 1 continued

females have decreased total weight at PND 16 ($t_{11}$ = 2.84, p=0.015) 21 ($t_{13}$ = 4.99, p<0.001) and 28 ($t_{14}$ = 5.73, p<0.001), but not at PND 35 ($t_{13}$ = 1.90, p=0.078). When compared to ctrl males, LB males have decreased total weight at PND 21 ($t_{17}$ = 6.64, p<0.001) and 28 ($t_{14}$ = 2.65, p=0.018), but not at 16 ($t_{18}$ = 1.90, p=0.073) or 35 ($t_{15}$ = 0.98, p=0.33). (Females: Ctrl n = 6, 8, 8, 7; LB n = 7, 7, 8, 8) (Males: Ctrl n = 10, 10, 8, 9; LB n = 10, 9, 8, 8). (C) Graphs depicting LB changes in total brain weight for females (top) and males (bottom). Insets show a close up of mean and SEM for PND 16 data. However, when we assessed brain weight, LB female mice did not differ from ctrl females at PND 16 ($t_9$ = 2.11, p=0.063), with differences emerging at PND 21 ($t_{13}$ = 5.62, p<0.001), 28 ($t_{14}$ = 3.11, p=0.0075), and 35 ($t_{12}$ = 3.55, p=0.0039). LB males had significantly different brain weight when compare to ctrl males at PND 16 ($t_{13}$ = 2.29, p=0.038), 21 ($t_{17}$ = 4.84, p<0.001) and 28 ($t_{14}$ = 3.13, p=0.0073), but not at PND 35 ($t_{14}$ = 0.40, p=0.69). (D) When we assessed brain to total weight ratio, female LB mice had a higher ratio at PND 16 ($t_9$ = 3.59, p=0.0057), 21 ($t_{13}$ = 3.69, p=0.0027), and 28 ($t_{14}$ = 4.33, p<0.001), but not at PND 35 ($t_{12}$ = 0.69, p=0.50) when compared to ctrl females. However, LB males did not significantly differ from ctrl males at PND 16 ($t_{13}$ = 0.76, p=0.45), however, by PND 21 ($t_{17}$ = 4.44, p<0.001) and 28 ($t_{14}$ = 2.60, p=0.020) LB males had significantly higher brain/total weight ratio when compared to ctrl males, with effects dissipating by PND 35 ($t_{14}$ = 0.51, p=0.61). For panels (C) and (D): (Females: Ctrl n = 6, 8, 8, 6; LB n = 5, 7, 8, 8) (Males: Ctrl n = 6, 10, 8, 8; LB n = 9, 9, 8, 8). Dots in panels represent individual data points. Bars represent group means + / - SEM. Unpaired two-tailed student t-tests were used for data analysis. *=p < 0.05, **=p < 0.01, ***=p < 0.001.

## LB is associated with deficits in fear recall during the peri-weaning period

To determine if LB rearing affected learning and expression of a conditioned fear memory, mice underwent fear conditioning training and testing. The age (in days) at which each separate cohort of mice underwent each stage of auditory fear learning is shown in *Figure 2A*. To dissociate cue-memory from context-memory, mice were habituated to two contexts (context A and B) for 5 min per context for 2 consecutive days. Mice were then conditioned to six tones (75 dB, 30 s) that each co-terminated with a foot-shock (0.57 mA, 1 s) in context A and tested for memory recall 24 hr post-conditioning in context B (*Figure 2B*). In control mice, we observed a significant developmental increase in freezing behavior from PND 19 to PND 22. However, in LB reared mice, we found significantly lower levels of freezing compared to control mice during cue recall at PND 22, but not at later developmental time points (*Figure 2C*), indicating a possible delay in cue-associated fear learning in LB mice. Further, a significant decrease in freezing to the conditioning tones was observed in LB males at PND 21 and in LB females at PND 28. To determine if differences in recall at PND 22 were the result of diminished learning at PND 21, conditioning curves of control and LB mice were matched for levels of freezing during acquisition (*Figure 2—figure supplement 1*). Matching was conducted by mixing male and female data, and systematically removing individual animal data from the analysis until LB and Ctrl conditioning curves at PND 21 were overlapping and not significantly different. The results showed that deficits in fear expression could not be solely explained by differences in freezing observed during conditioning. To determine if the lower levels of freezing in LB mice at PND 22 were the result of deficits in acquisition, consolidation, or recall; separate cohorts of male and female mice were conditioned at PND 21 and tested at either 1 hr (pre-consolidation), 6 hr (post-consolidation), 24 hr (short-term recall), or 7 days (long-term recall) post conditioning (*Figure 3A*). At 1 hr, there was a small but significant decrease in freezing observed in LB reared mice compared to controls. At 6 hr, no differences in levels of freezing were detected between LB and control reared mice, indicating that LB mice could freeze to the cue and appeared to have consolidated the fear memory. By 24 hr, LB mice showed significantly lower freezing compared to control mice, suggestive of deficits in fear memory. Interestingly, at 7 days post conditioning, LB freezing levels did not differ from control levels. In aggregate, initial learning was intact but freezing to the conditioned tones was transiently impaired 24 hr post conditioning. Together the results suggest that LB reared mice form a cue-associated fear memory at PND 21 but are not able to behaviorally express this memory during early development. However, with time, behavioral expression of fear, in the form of freezing, re-emerges. Follow-up analysis on sex differences were performed (*Figure 3B*) showing that both LB males and LB females showed the decrease in freezing 24 hr post condition followed by an increase in freezing 7 days post conditioning. A significant decrease in freezing was observed 1 hr post conditioning in control females when compared to control males. A battery of tests were conducted in males and females to ensure that effects of fear expression were not due to differences in anxiety-like behavior (light/dark box test: *Figure 3C* and *Figure 3—figure supplement 1*; elevated plus maze: *Figure 3—figure supplement 2*), locomotion (open field test: *Figure 3—figure supplement 3*), or foot-shock sensitivity and reactivity (*Figure 3—figure*

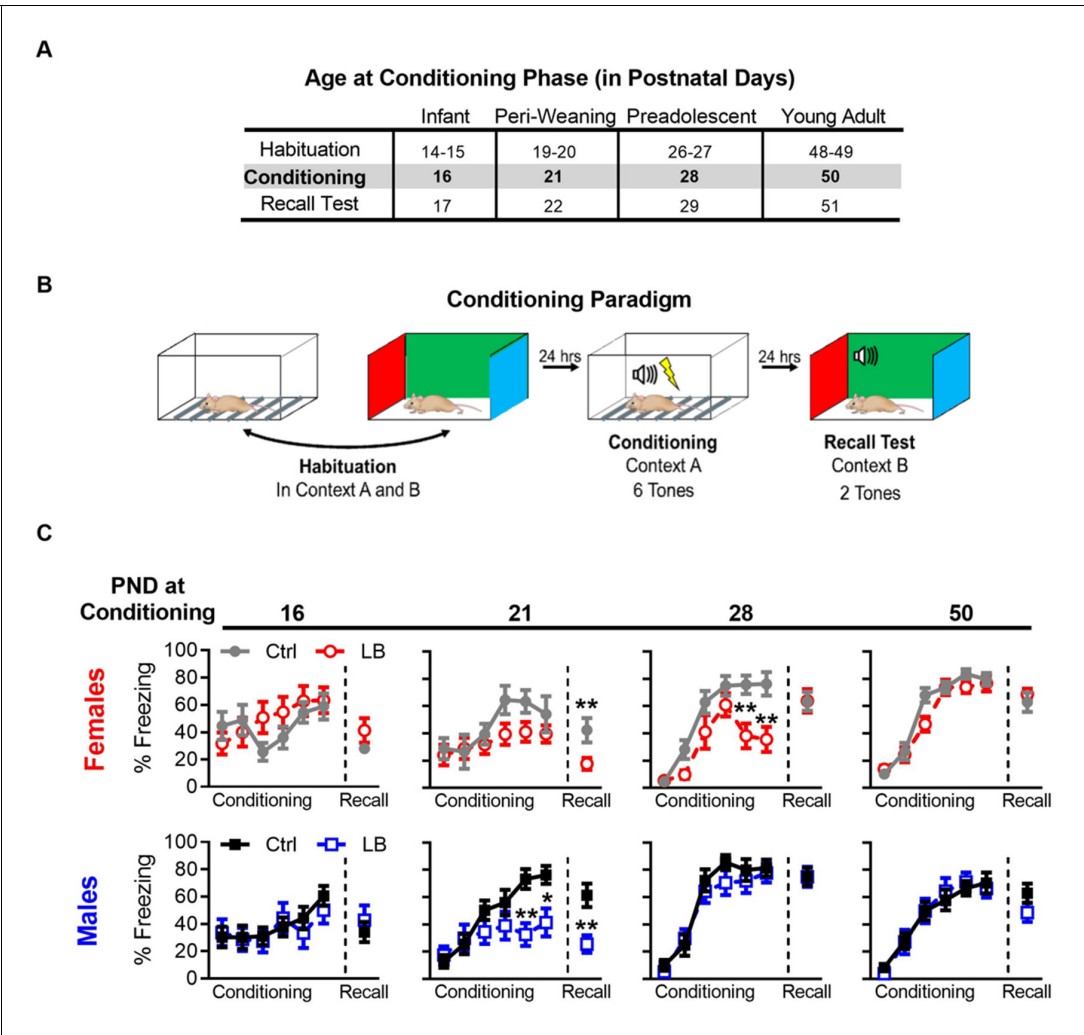

**Figure 2.** LB affects the short-term expression of fear during early development. (**A**) Table of the postnatal day at which each cohort underwent different phases of auditory cue conditioning. Different cohorts were used for each age group. (**B**) Schematic of auditory fear conditioning protocol. (**C**) Graphs of the percent time female (top) and male (bottom) mice spent freezing (immobile) during the conditioning and recall test. LB female ($t_{21}$ = 2.79, p=0.010) and male ($t_{20}$ = 3.39, p=0.0028) mice exhibited decreased fear expression at PND 22 when compared to age and sex matched controls, an effect not observed at other ages (Females: PND 17 ($t_{20}$ = 1.43, p=0.16), 29 ($t_{19}$ = 0.032, p=0.97), 51 ($t_{21}$ = 0.73, p=0.47)) (Males: PND 17 ($t_{24}$ = 0.68, p=0.50), 29 ($t_{21}$ = 0.014, p=0.98), 51 ($t_{19}$ = 1.42, p=0.16)). During auditory cue conditioning in females, a main effect of rearing condition was observed at PND 28 ($F_{(1,19)}$ = 7.70, p=0.012), but not at 16 ($F_{(1,20)}$ = 0.52, p=0.47), 21 ($F_{(1,21)}$ = 3.32, p=0.082), or 50 ($F_{(1,21)}$ = 1.33, p=0.26). Post-hoc analysis revealed that PND 28 LB females had significantly lower levels of freezing on the 5th ($t_{114}$ = 3.50, p=0.0039) and 6th ($t_{114}$ = 3.80, p=0.0014) tones. A main effect of tone during conditioning was observed in females at all ages (PND 16: $F_{(5,100)}$ = 2.70, p=0.024; PND 21: $F_{(5,105)}$ = 4.11, p=0.0019; PND 28: $F_{(5,95)}$ = 39.3, p<0.0001; PND 50: $F_{(5,105)}$ = 104.1, p<0.0001), indicating that female mice learned the tone/foot shock association at all ages tested. No interaction between tone trial and rearing condition where observed at PND 16 ($F_{(5,100)}$ = 1.51, p=0.19) or 21 ($F_{(5,105)}$ = 0.83, p=0.52); however, significant interactions where observed at PND 28 ($F_{(5,95)}$ = 3.79, p=0.0035) and 50 ($F_{(5,105)}$ = 2.45, p=0.037). During auditory cue conditioning in males, no main effect of rearing condition was observed at any age tested (PND 16: $F_{(1,24)}$ = 0.15, p=0.69; PND 21: $F_{(1,20)}$ = 3.32, p=0.083; PND 28: $F_{(1,21)}$ = 0.78, p=0.38; PND 50: $F_{(1,19)}$ = 0.0017, p=0.96). However, a post-hoc analysis revealed that PND 21 LB males had significantly lower levels of freezing on the 5th ($t_{120}$ = 3.25, p=0.0088) and 6th ($t_{120}$ = 2.78, p=0.036) tones when compared to ctrl males. During conditioning in males a main effect of tone trial was observed at all ages (PND 16: $F_{(5,120)}$ = 3.33, p=0.0074; PND 21: $F_{(5,100)}$ = 12.14, p<0.0001; PND 28: $F_{(5,105)}$ = 54.63, p<0.0001; PND 50: $F_{(5,95)}$ = 62.84, p<0.0001), indicating that male mice learned the tone/foot shock association at all ages tested. No interaction between tone trial and rearing condition where observed in male conditioning at PND 16 ($F_{(5,120)}$ = 0.42, p=0.83), 28 ($F_{(5,105)}$ = 0.59, p=0.70), or 50 ($F_{(5,95)}$ = 0.49, p=0.77); however, a significant interaction was observed at PND 21 ($F_{(5,100)}$ = 4.05, p=0.0022). (Females: Ctrl n = 11, 8, 12, 11; LB n = 11, 15, 9, 12) (Males: Ctrl n = 16, 9, 12, 12; LB n = 10, 13, 11, 12). For effects on memory recall, following matching for conditioning curves see *Figure 3—figure supplement 4*. Additional tests for somatosensation were conducted to ensure that differences in freezing were not due to differences in foot-shock sensitivity, see Figure 2—figure supplement 2. Bars represent group means + / - SEM. Two-way repeated measure ANOVA followed by a Sidak's multiple comparison analysis was used to analyze the conditioning curves. Unpaired two-tailed student t-tests were used to analyze differences in recall tests. *=p < 0.05, **=p < 0.01, ***=p < 0.001.

*Figure 2 continued on next page*

*Figure 2 continued*

The online version of this article includes the following figure supplement(s) for figure 2:

**Figure supplement 1.** Effects on recall at PND 21 are not a consequence of conditioning deficits.

*supplement 4*). No general differences were detected between LB and control at PND 21 in anxiety-like behavior, locomotion, or somatosensation (see supplements for full stats).

## LB increased PV-positive cell density early in life in the BLA, but not mPFC

The altered trajectory of fear expression could have arisen from developmental changes in circuits supporting this behavior, including mPFC and BLA, which are known to be involved in emotional regulation and fear conditioning (*Arruda-Carvalho and Clem, 2015*; *Etkin et al., 2011*; *Sierra-Mercado et al., 2011*). Our prior work demonstrated that LB rearing accelerates markers of development, including PV interneuron maturation, in the hippocampus (*Goodwill et al., 2018*). Therefore, we predicted that deficits in freezing could be due to ELA effects on interneuron maturation in either the BLA or mPFC. To test this hypothesis, we tested for developmental changes in markers of subclasses of interneurons across these distinct brain structures.

Using immunohistochemistry, we labeled PV-positive interneurons, a late differentiating subclass of inhibitory interneurons (*Bartolini et al., 2013*; *Mukhopadhyay et al., 2009*; *Rymar and Sadikot, 2007*), in the brain of LB and control reared male mice at PND 16, 21, 28, 50, and 75. PV-positive interneurons are known to begin differentiating between PND 10–28, reaching a mature phenotype at approximately PND 30 (*Berdel and Moryś, 2000*; *Dávila et al., 2008*). In BLA, LB led to an increased PV-positive cell density at PND 21 when compared to control reared mice at this age (*Figure 4A*). No differences in PV-positive cell density was observed at PND 16, with both groups expressing low densities of PV-positive cells. Furthermore, the difference in PV-positive cell density between LB and control groups subsided by PND 28, with control PV-positive cell density rising to match those observed in LB mice. The increase in PV-positive cell density in the BLA of LB mice at PND 21 was not observed in the other regions of the fear circuit assessed here, including the prelimbic (PL; *Figure 4A* center) and infralimbic (IL; *Figure 4A* right) subregions of the mPFC. A non-significant decrease in PV-positive cell density in LB mice was observed at PND 50 in BLA, IL, and PL. This decrease in PV-positive cell density was not observed in other brain regions such as the primary motor cortex and somatosensory cortex (*Figure 4—figure supplement 1*). Thus, the decrease in PV-positive cell density may be indicative of changes in parvalbumin protein concentration within BLA, IL, and PL as mice transition into adulthood. Studies assessing protein concentrations in combination with additional labeling techniques will be needed to fully determine whether this represents a true decline. Together these findings suggest that LB increased PV positive cell density in the BLA but not the mPFC, suggesting region-specific effects of LB rearing.

Next, western blot analysis was used to determine if LB impacted protein levels for other classes of interneurons (calbindin/calretinin), glutamatergic neuronal markers (VGLUT1), or markers of myelination (myelin basic protein) across early developmental time points in male mice (*Figure 4B and C*). Western blot analysis allowed us to detect differences between LB and Ctrl mice that could not be detected through immunohistochemical labeling due to poor quality of staining in brains from young animals. In BLA, LB had no effect on calretinin, calbindin, VGLUT1, or myelin basic protein levels at any age tested (*Figure 4B*). Furthermore, in mPFC, LB did not affect levels of calretinin, calbindin, or VGLUT 1. However, LB was associated with increased myelin basic protein levels in mPFC at PND 28 (*Figure 4C*). Together, LB appeared to have driven a cell-type-specific developmental increase in PV-positive cells in BLA.

## PV inhibition in BLA rescues the fear expression deficit

We then hypothesized that the precocious maturation of PV-positive cells in BLA may have impacted the expression of freezing behavior at PND 22. The focus on PV-positive cells was based on previous reports that have demonstrated a role for BLA PV-positive cells in modulating fear expression (*Davis et al., 2017*; *Wolff et al., 2014*). To test this hypothesis, we used transgenic mice that selectively express the optogenetic construct halorhodopsin in PV-positive cells. Mice that express

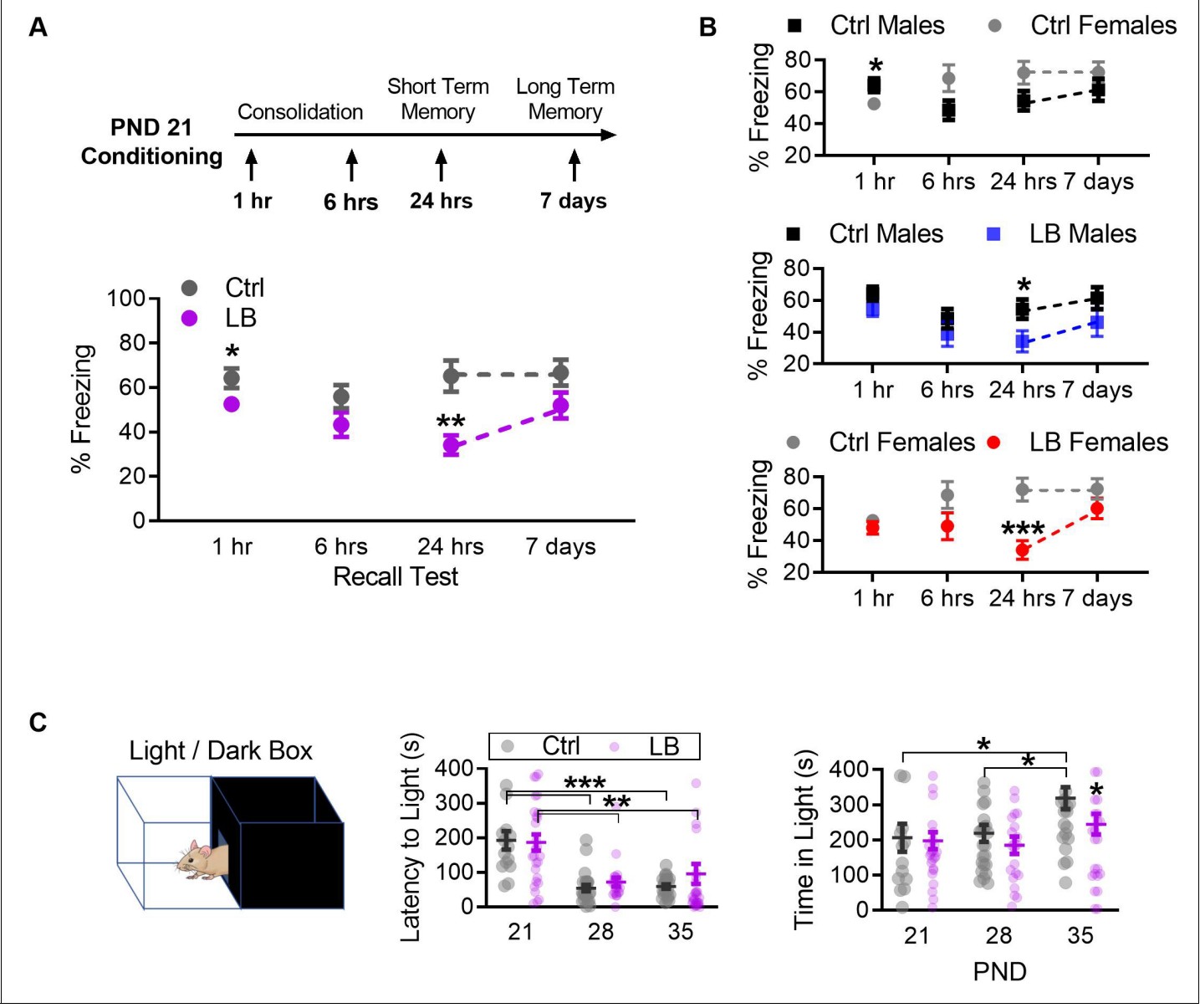

**Figure 3.** LB affects the short-term expression of fear during early development. (**A**) Schematic of experimental protocol (top). A mix of male and female mice were conditioned at PND 21 and tested at only one time point: 1 hr, 6 hr, 7 hr, or 7 days post-conditioning. Graph depicting changes in freezing levels of distinct cohorts of mice during recall tests at varying delays (bottom). LB mice had decreased freezing at 1 hr ($t_{74}$ = 2.11, p=0.037), 24 hr ($t_{48}$ = 3.41, p=0.0012) but not 6 hr ($t_{55}$ = 1.66, p=0.10) or 7 days ($t_{46}$ = 1.76, p=0.085) post-conditioning. (Ctrl n = 34, 30, 29, 27; LB n = 42, 27, 21, 21). (**B**) Re-analysis of data presented in panel A to reveal sex differences within the data. When compared to Ctrl males, Ctrl females had decreased freezing at 1 hr ($t_{32}$ = 2.11, p=0.042), but not at 6 hr ($t_{28}$ = 1.95, p=0.060), 24 hr ($t_{27}$ = 1.89, p=0.069) or 7 days ($t_{25}$ = 1.17, p=0.25) post-conditioning. LB males ($t_{25}$ = 2.21, p=0.036) and LB females ($t_{21}$ = 3.94, p=0.0007) had decreased freezing 24 hr post-conditioning when compared to sex matched controls. No other significant differences were observed between Ctrl males and LB males (1 hr: $t_{42}$ = 1.32, p=0.19; 6 hr: $t_{32}$ = 0.99, p=0.32; 7 days: $t_{23}$ = 1.37, p=0.18), or between Ctrl females and LB females (1 hr: $t_{30}$ = 0.85, p=0.39; 6 hr: $t_{21}$ = 1.64, p=0.11; 7 days: $t_{21}$ = 1.31, p=0.20). (Ctrl males n = 15, 19, 16, 14; Ctrl females n = 19, 11, 13, 13; LB males n = 29, 15, 11, 11; LB females n = 13, 12, 10, 10). (**C**) Depiction of the light/dark box used to assess anxiety-like behavior (top). A mix of male and female mice were placed in the dark side of the box, the latency to light (center) and the total time spent in the light side of the box (bottom) are shown. Total time spent in the light/dark box was 420 s. Age ($F_{(2,135)}$ = 19.2, p<0.0001), but not rearing condition ($F_{(1,135)}$ = 0.50, p=0.47) or age x rearing interaction ($F_{(2,135)}$ = 0.72, p=0.48), significantly affected the latency to exit the dark side of the box (center). PND 21 mice of both LB and control reared conditions took more time to enter the light side of the box when compared to mice from the same rearing condition at PND 28 (LB $t_{135}$ = 3.63, p=0.0012; Ctrl $t_{135}$ = 4.05, p=0.0003) and PND 35 (LB $t_{135}$ = 3.09, p=0.0073; Ctrl $t_{135}$ = 3.89, p=0.0005). However, a main effect of rearing condition on the total time mice spent in the light side of the light/dark box ($F_{(2,135)}$ = 3.91, p=0.022) was observed (bottom). A post-hoc analysis revealed that significant differences in rearing condition were only observed at PND 35 ($t_{135}$ = 2.60, p=0.030).

*Figure 3 continued on next page*

*Figure 3 continued*

Furthermore, a significant main effect of age on the time spent in the light side ($F_{(1,135)} = 4.01$, p=0.047) was observed. Specifically, control PND 35 mice spent significantly more time in the light side when compared to control reared mice aged PND 21 ($t_{135} = 2.46$, p=0.043) and PND 28 ($t_{135} = 2.56$, p=0.033). No interaction between age and rearing condition was observed ($F_{(2,135)} = 1.23$, p=0.29). (Ctrl n = 14, 24, 23; LB n = 27, 22, 31). For effects of rearing condition by sex see *Figure 3—figure supplement 1*. Bars represent group means + / - SEM. Dots in panel (**C**) represent individual data points. Unpaired two-tailed student t-tests were used in (**A**) and (**B**). For (**C**) a two-way ANOVA followed by a Sidak's multiple comparison analysis was used. \*=$p < 0.05$, \*\*=$p < 0.01$, \*\*\*=$p < 0.001$.

The online version of this article includes the following figure supplement(s) for figure 3:

**Figure supplement 1.** LB did not affect anxiety like behavior at PND 21.
**Figure supplement 2.** LB mice spent more time performing head dips in the elevated plus maze (EPM) at PND 21.
**Figure supplement 3.** LB did not affect locomotion at PND 21.
**Figure supplement 4.** LB did not affect somatosensation at PND 21.

halorhodopsin were also positive for EGFP, allowing us to verify ELA effects on cell density. This strategy allowed us to silence PV-positive cells in a time and region-specific manner.

We first replicated the ability of ELA to increase PV-positive cell density in the BLA by quantifying EGFP-positive cells in control and LB reared mice from the PV Halo mice (*PV-Cre*[Het]/*floxed NpHR*[Het]), which co-express an EGFP reporter on PV-positive cells. Again, LB was associated with an increase in BLA PV-positive cell density at PND 21 mirroring prior immunohistochemical findings (*Figure 5A*). Specifically, at PND 21, LB PV Halo mice had a greater density of PV positive cells in the BLA compared to age matched control PV Halo mice (*Figure 5A*), with no effect of LB rearing on PV-positive cell density in PL or IL in this mouse line (*Figure 5B*). No effect of LB rearing at PND 21 was observed for other genetically labeled cell populations within the BLA, such as somatostatin and VGlut 2 (*Figure 5—figure supplement 1*).

To test if inhibiting PV-positive cells in the BLA of LB reared mice during fear learning could rescue the observed freezing deficits at PND 22, we optogenetically silenced PV positive cells in the BLA during fear conditioning. Consistent with our prediction, bilateral inactivation of PV-positive cells in the BLA during the conditioning tones resulted in increased freezing 24 hr later, during the recall test (*Figure 5C*). To ensure that the increased freezing was not due to effects on locomotion or anxiety-like behavior, mice were placed in an open field and PV-positive cells in the BLA were optogenetically inhibited and the behavior of mice was tracked. No effects of optogenetic inhibition of PV-positive cells were found for measures of anxiety-like behavior or general locomotor activity of mice (*Figure 5—figure supplement 2A*). To ensure that inhibition of PV-positive cells was modulating BLA activity, a subset of unilaterally implanted mice were administered light for 15 min while freely moving in their homecage. Since parvalbumin neurons work to inhibit neuronal activity, we hypothesized that inactivation of the inhibitory interneurons should lead to increased activity of excitatory neurons and therefore increased expression of the immediate early gene cFos. We found that mice exhibited increased cFos labeling in the optogenetically inhibited side when compared to the non-inhibited (control) side (*Figure 5—figure supplement 2B*). Further, electrophysiological control experiments carried out by our lab in the OFC of the same mouse line demonstrated robust inhibition of PV-positive cells in response to light (*Goodwill et al., 2018*). Together, these results suggest that the inhibition of PV-positive cells in the BLA of LB reared mice is sufficient to overcome LB induced freezing deficits at this age.

## LB rearing increased PL to BLA anatomical connectivity in females

Previous research has demonstrated that ELA can alter mPFC to BLA connectivity in humans (*Fan et al., 2014*; *Gee et al., 2013*; *Herringa et al., 2016*) (reviewed in *Herzberg and Gunnar, 2020*; *VanTieghem and Tottenham, 2018*), and BLA to mPFC anatomical (*Honeycutt et al., 2020*), and functional (*Bolton et al., 2018*; *Guadagno et al., 2018*; *Yan et al., 2017*) connectivity in rodents. Since LB male and female mice exhibited diminished fear expression at PND 21, and PL projections into BLA are known to promote fear learning, while IL projections into BLA support fear extinction (*Do-Monte et al., 2015a*; *Giustino and Maren, 2015*; *Lee and Choi, 2012*; *Sierra-Mercado et al., 2011*), we sought to investigate whether LB altered anatomical connectivity between mPFC and BLA in our hands. To test if LB altered the timing and density of projections from mPFC to BLA, the retrograde tracer cholera toxin B (CTB) was injected unilaterally into the BLA

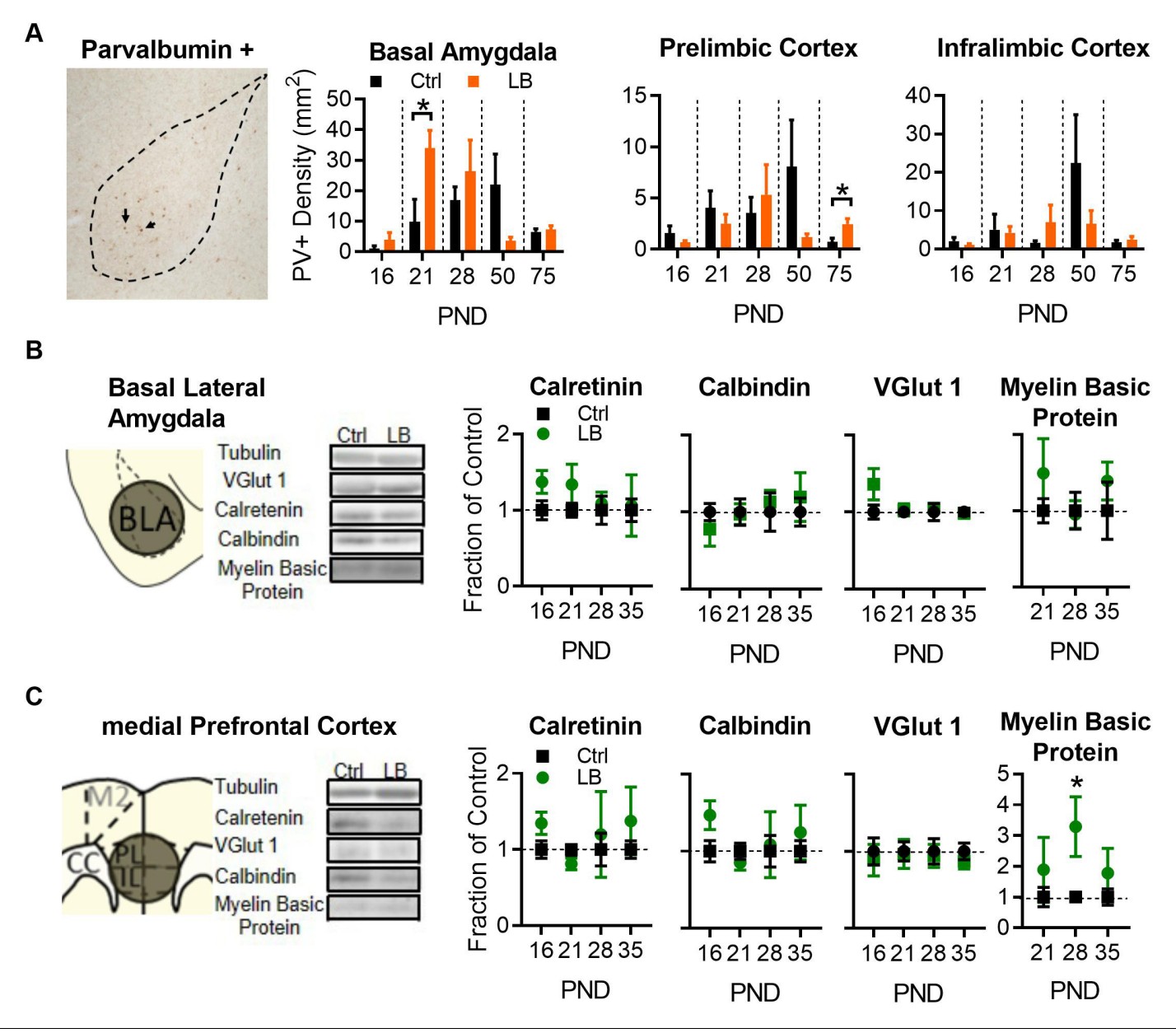

**Figure 4.** LB alters select markers for neuronal populations in mPFC and BLA. (**A**) Representative image of immunohistochemical labeling in the BLA of an LB PND 21 mouse (left). From left to right, graphs showing the density of PV-positive cells in the BLA, PL, and IL of male mice at different ages. Immunohistochemical analysis revealed that LB significantly increased Parvalbumin positive (PV+) neurons at PND 21 in BLA ($t_{10} = 2.40$, p=0.037) but not in PL ($t_{10} = 0.83$, p=0.42) or IL ($t_{10} = 0.17$, p=0.86). Except for PL at PND 75 ($t_9 = 2.85$, p=0.018), no differences were observed at PND 16, 28, 50 or 75 respectively for BLA ($t_8 = 1.07$, p=0.31; $t_8 = 0.85$, p=0.41; $t_9 = 1.65$, p=0.13; $t_9 = 0.48$, p=0.63), PL ($t_8 = 1.26$, p=0.24; $t_{10} = 0.54$, p=0.59; $t_{11} = 1.40$, p=0.18; $t_9 = 2.85$, p=0.018), or IL ($t_8 = 1.03$, p=0.33; $t_{10} = 1.15$, p=0.27; $t_{11} = 1.12$, p=0.28; $t_9 = 0.81$, p=0.43). For BLA (Ctrl n = 5, 7, 5, 6, 6; LB n = 5, 5, 5, 5, 5). For PL and IL (Ctrl n = 5, 6,6,7,6; LB n = 5, 6, 6, 6, 5). (**B**) Western blot analysis of BLA tissue. Diagram and sample blots of PND 35 mice are shown. Graphs showing differing protein levels of mice from infancy into adolescence. No differences between LB and control mice were observed in BLA for calretinin ($t_{11} = 1.90$, p=0.083; $t_{11} = 1.28$, p=0.22; $t_{11} = 0.37$, p=0.71; $t_{12} = 0.14$, p=0.88), calbindin ($t_{11} = 0.94$, p=0.36; $t_{11} = 0.12$, p=0.90; $t_{11} = 0.46$, p=0.65; $t_{12} = 0.53$, p=0.60), VGLUT1 ($t_{11} = 1.53$, p=0.15; $t_{12} = 0.26$, p=0.79; $t_{12} = 0.41$, p=0.68; $t_{12} = 0.18$, p=0.85), or myelin basic protein ($t_{11} = 1.08$, p=0.30; $t_{11} = 0.16$, p=0.87; $t_{12} = 0.86$, p=0.40) levels. For calbindin, calretinin, and myelin basic protein (Ctrl n = 7, 7, 6, 7; LB n = 6, 6, 7, 7). For VGLUT1 (Ctrl n = 6, 7, 7, 7; LB n = 7, 7, 7, 7). (**C**) Western blot analysis of mPFC tissue. Diagram and sample blots of PND 35 mice are shown. Graphs showing differing protein levels of male mice from infancy into adolescence. No differences between LB and control mice were observed in mPFC for calretinin ($t_{10} = 1.84$, p=0.095, $t_{12} = 1.71$, p=0.11, $t_9 = 0.35$, p=0.73, $t_{11} = 0.88$, p=0.39), calbindin ($t_{10} = 2.01$, p=0.071; $t_{12} = 1.02$, p=0.32; $t_9 = 0.19$, p=0.85; $t_{11} = 0.67$, p=0.51), or VGLUT1 ($t_{10} = 0.42$, p=0.68; $t_{12} = 0.15$, p=0.88; $t_{12} = 0.26$, p=0.79; $t_{12} = 0.85$, p=0.41) levels. An increase in mPFC Myelin Basic Protein was observed exclusively at PND 28 ($t_9 = 2.54$, p=0.031), but not at PND 21 ($t_{12} = 0.80$, p=0.43) or 35 ($t_{11} = 0.97$, p=0.35). For calbindin,

*Figure 4 continued on next page*

*Figure 4 continued*

calretinin, and myelin basic protein (Ctrl n = 6, 7, 6, 7; LB n = 6, 7, 5, 6). No effects of VGLUT1 were observed in the mPFC. For VGLUT1 (Ctrl n = 6, 7, 7, 7; LB n = 6, 7, 7, 7). Bars represent group means + / - SEM. Unpaired two-tailed t-tests between control and LB of a given age were used for A, B and C. *=*p* < 0.05, **=*p* < 0.01, ***=*p* < 0.001.

The online version of this article includes the following figure supplement(s) for figure 4:

**Figure supplement 1.** Density of Immunohistochemically stained parvalbumin-positive cells (PV+) in the rostral primary motor cortex and the rostral primary somatosensory cortex of male mice.

and the number of labeled cells in PL and IL were quantified. Injections were performed 1 day prior to the time point of interest (e.g. PND 15 for PND 16) and perfusions performed 1 day later (e.g. PND 17 for PND 16). Four time points were tested (PND 16, 21, 28 and 35; *Figure 6A–C*). As labeling could be affected by efficiency of CTB uptake following injection and BLA placement, we used an ANCOVA analysis with BLA area at the site of injection and CTB injection area as covariates. Previous research has reported no effects of LB rearing on BLA volume at PND 20 (*Guadagno et al., 2018*). Therefore, we interpreted the corrected estimated mean densities, obtained through the ANCOVA analysis as accounting for between animal differences in CTB injection size and injection placement (potential limitations of this approach may be found in the Discussion and Materials and methods). An independent ANCOVA was conducted for each age group and each sex (for analysis of Ctrl males vs Ctrl females see *Figure 6—figure supplement 1*). In males, LB did not affect the density of PL to BLA or IL to BLA labeled cells at PND 16, 21, 28 or 35. However, LB female mice did show an increase in PL to BLA projecting cells at PND 21 and 28 when compared to Ctrl females (*Figure 6D and E*). LB females did not differ from Ctrl females in IL to BLA projection densities. Next, we sought to determine if LB altered the balance of PL and IL inputs into BLA across development. We determined the relative difference in PL and IL projections as a factor of the total number of labelled cells for each mouse ((PL - IL) / (PL + IL)). This value, or projection index, accounts for between-subject differences in labeling efficiency (*Figure 6F*). No differences in projection index were found between LB and Ctrl reared male mice. A significant increase in the projection index was observed in LB reared females when compared to Ctrl females at PND 16, but not at PND 21, 28 or 35, suggesting an upwards shift in PL over IL inputs into BLA at that age in females. These results suggest that females may be more sensitive to LB induced alterations in the development of mPFC anatomical connectivity to BLA.

## Discussion

Here, we identify LB effects on the developmental expression of cued fear learning and provide a potential causal link between the neurodevelopmental and behavioral consequences of LB rearing on fear learning and developmental fear expression. We found that LB decreased physical growth during postnatal development, with a more robust effect on the body than the brain. Delayed brain growth was accompanied by significant effects on the timing of maturation (as indexed by developmental change in cell density or protein levels) for specific cell populations in select brain regions. LB led to a selective increase in PV-positive cell density within the BLA at PND 21 (peri-weaning). Furthermore, LB reared mice at PND 21 were able to learn, but not express, an auditory conditioned fear response, which was rescued by inhibiting PV-positive cell activity in the BLA of LB reared mice. In addition, LB rearing led to sex-specific changes in the anatomical connectivity of PL and BLA in female peri-adolescent mice. Together this data supports the idea that LB rearing alters the timing of maturation of subclasses of cells in key brain regions supporting threat learning and the expression of fear memories, with possible implications for later pathology development.

Previous research in humans has shown that childhood resource insecurity in the form of poverty, specifically during infancy, is associated with altered regional brain volume, with the prefrontal cortex and amygdala being affected (*Hair et al., 2015*; *Hanson et al., 2013*; *Luby et al., 2013*; *Sheridan et al., 2012*). However, the mechanisms by which poverty confers risk for altered brain development and how these factors impact the development of region and cell specific populations of neurons remains largely unknown. Here, rearing mice under conditions of resource restriction, altered the development of select populations of neurons in the mPFC and BLA, key regions involved in fear learning and threat assessment. In the BLA, LB led to an earlier rise in the density of

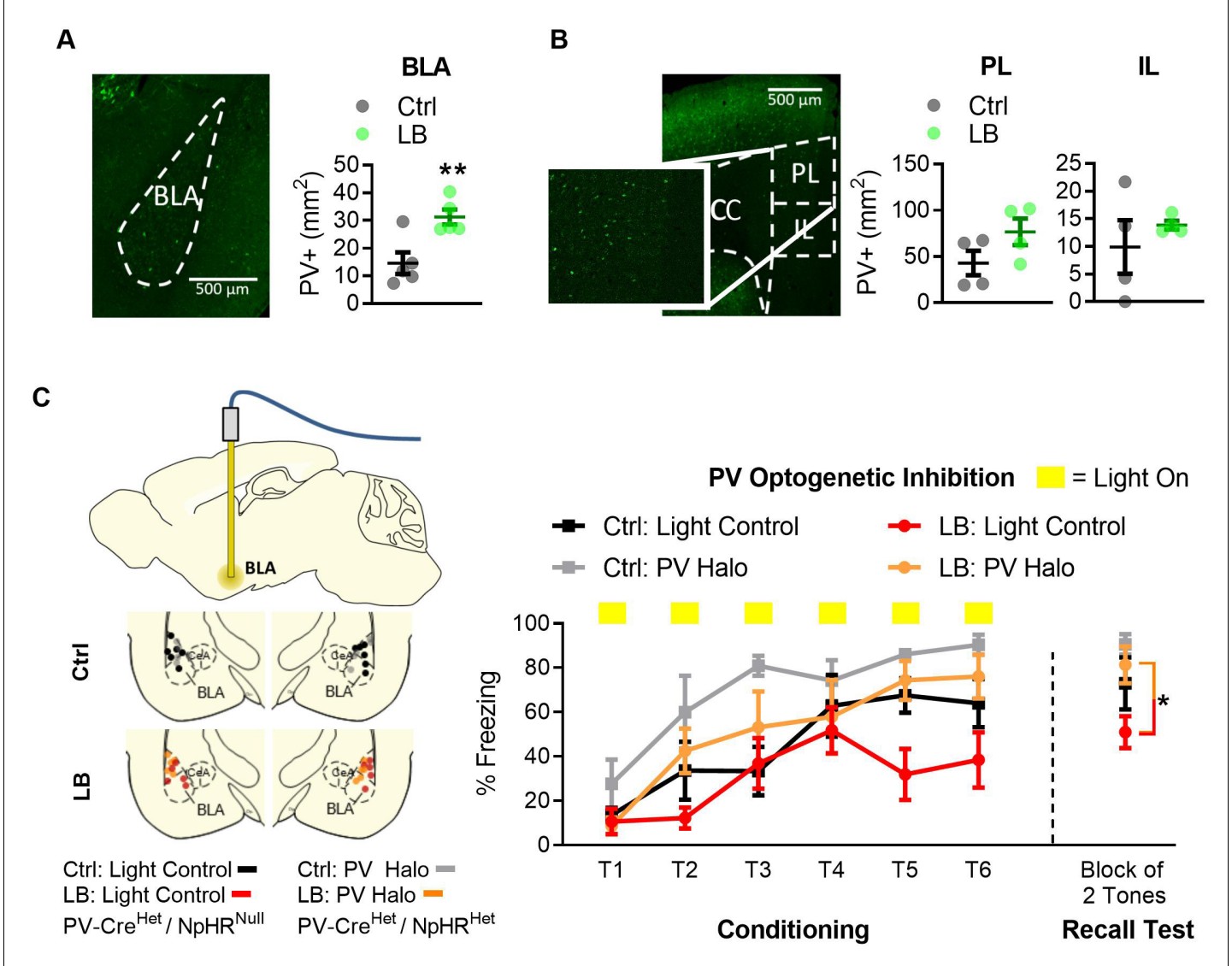

**Figure 5.** Optogenetic inactivation of parvalbumin-positive (PV+) cells was sufficient to increase fear expression at PND 22 in LB mice. (**A**) Representative image of PV-positive cells in BLA of Het PV-Cre/Het floxed halo ('PV Halo') mouse line (left). LB increased the density of PV-positive cells in the BLA ($t_8$ = 3.49, p=0.0081; n = 5 per group). (**B**) Representative image of PV+ cells in the mPFC of a PV Halo (left). LB did not affect the density of PV-positive cells in PL ($t_6$ = 1.72, p=0.13) or IL ($t_6$ = 0.80, p=0.45) compared with control reared mice (n = 4 per group). (**C**) Graph showing conditioning and recall test. Optogenetic inhibition occurred during the conditioning tones. Ctrl and LB reared Light control (Het PV-Cre/null floxed Halo) and PV Halo (Het *PV-Cre*/Het floxed Halo) mice were bilaterally implanted with an optical fiber into BLA, optic fiber placements are shown (left). To assess effects of optogenetic inhibition of PV cells in LB reared mice we compared LB light controls versus LB PV Halo mice. When analyzing PV inhibition in LB mice, a main effect of tone presentation ($F_{(5,45)}$ = 6.97, p<0.0001) was observed, indicating mice had learned the tone/foot shock association. Optogenetic inactivation of PV positive cells in LB reared mice increased freezing during conditioning ($F_{(1,9)}$ = 6.21, p=0.034), and resulted in an increase in freezing on the recall test 24 hr later ($t_9$ = 2.69, p=0.024). No interaction between tone presentation and experimental condition was observed ($F_{(5,45)}$ = 1.55, p=0.19) between LB light control and LB PV Halo mice. For LB: light control n = 5, for LB: PV Halo n = 6. When assessing the effects of PV-positive cell inhibition between Ctrl: Light control and Ctrl: PV Halo mice, a repeated measure ANOVA revealed a main effect of tone presentation ($F_{(5,45)}$ = 9.08, p<0.0001), and optogenetic inactivation ($F_{(1,9)}$ = 9.5, p=0.013). No main effect of tone by optogenetic inactivation ($F_{(5,45)}$ = 0.96, p=0.44) was observed. No differences in memory retrieval were observed between the Ctrl: light control and the Ctrl: PV Halo mice ($t_9$ = 1.29, p=0.22). For additional locomotion controls and cFos analysis of inhibition see *Figure 4—figure supplement 1*. A repeat measure two-way ANOVA was used to assess differences in fear conditioning, for all other analysis two-tailed unpaired student t-tests were used. Bars represent group means + / - SEM. *=*p* < 0.05, **=*p* < 0.01, ***=*p* < 0.001.

The online version of this article includes the following figure supplement(s) for figure 5:

**Figure supplement 1.** LB does not affect Somatostatin or VGlut2-positive cell densities in the BLA at PND 21.

**Figure supplement 2.** Optogenetic inhibition did not affect locomotion but did increase cFos in the BLA.

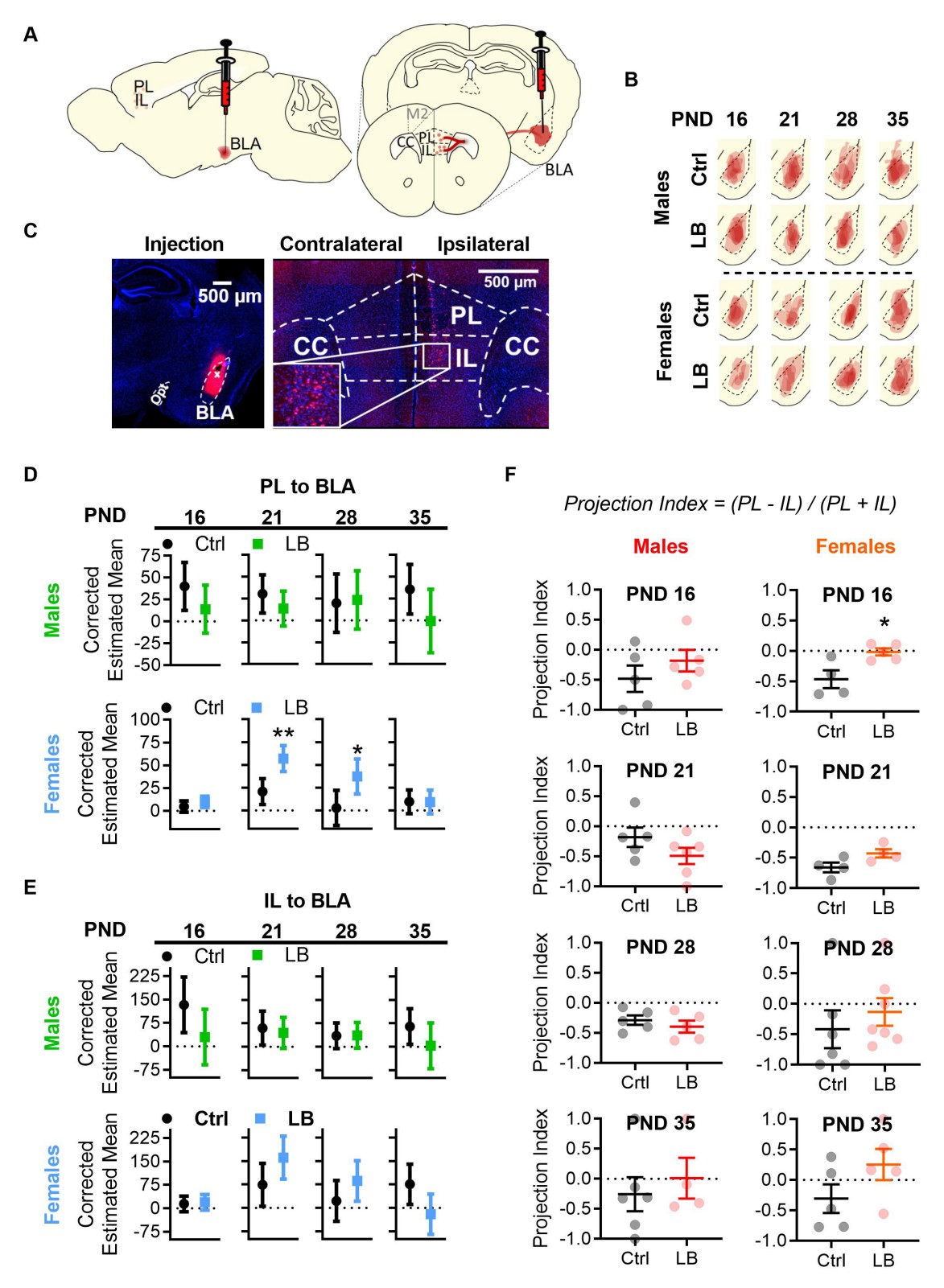

**Figure 6.** LB altered mPFC to BLA anatomical connectivity in females but not males. (**A**) Diagram of CTB 594 retrograde injection into the BLA. (**B**) Tracings of CTB injections into the BLA. (**C**) Example picture of injection in the BLA (left), and of cells labelled in mPFC (right). (**D**) Graphs showing the differences in PL to BLA corrected estimated mean densities with 95% confidence intervals from infancy (PND16) into adolescence (PND 35). In males, the density of PL to BLA projecting cells in LB mice did not differ at PND 16 ($F_{(1,6)}$ = 2.46, p=0.16), 21 ($F_{(1,7)}$ = 1.69, p=0.23), 28 ($F_{(1,6)}$ = 0.035, p=0.85) or

*Figure 6 continued on next page*

*Figure 6 continued*

35 ($F_{(1,6)}$ = 3.08, p=0.13) when compared to control mice. In females, a significant increase in PL to BLA projecting cells density was observed in LB mice at PND 21 ($F_{(1,4)}$ = 21.74, p=0.010) and 28 ($F_{(1,10)}$ = 7.44, p=0.021) but not at PND 16 ($F_{(1,6)}$ = 1.47, p=0.27) or 35 ($F_{(1,6)}$ = 0.001, p=0.97). (E) Graphs showing the differences in IL to BLA corrected estimated mean densities with 95% confidence intervals from infancy into adolescence. LB did not affect the density of IL to BLA projections at any age for males (16: $F_{(1,6)}$ = 3.71, p=0.10; 21: $F_{(1,7)}$ = 0.22, p=0.65; 28: $F_{(1,6)}$ = 0.003, p=0.958; 35: $F_{(1,6)}$ = 2.17, p=0.19) or females (16: $F_{(1,6)}$ = 0.091, p=0.77; 21: $F_{(1,4)}$ = 5.35, p=0.082; 28: $F_{(1,10)}$ = 2.212, p=0.168; 35: $F_{(1,6)}$ = 5.10, p=0.065). For panels D and E (Ctrl male n = 5, 5, 5, 6; LB male n = 5, 6, 5, 4; Ctrl female n = 5, 4, 7, 5; LB female n = 5, 4, 7, 5). (F) Graphs showing the differences in projection index from infancy into adolescence. LB altered the balance of PL and IL projections to BLA at PND 16 in females ($t_7$ = 3.09, p=0.0175). No other significant differences were observed in the projection index for males (16: $t_8$ = 1.05, p=0.32; 21: $t_9$ = 1.47, p=0.17; 28: $t_8$ = 0.83, p=0.42; 35: $t_8$ = 0.60, p=0.56) or females (21: $t_6$ = 2.22, p=0.068; 28: $t_{11}$ = 0.75, p=0.46; 35: $t_8$ = 1.61, p=0.14). (Ctrl male n = 5, 5, 5, 6; LB male n = 5, 6, 5, 4; Ctrl female n = 4, 4, 6, 5; LB female n = 5, 4, 7, 5). For D and E dots represent estimated group means with 95% confidence intervals. Estimated means were derived from ANCOVA analysis using two covariates: 1) the area of the CTB injection and 2) the area of BLA where the injection occurred. An ANCOVA was run per age per sex. Statistical significance was determined through post ANCOVA univariate comparison. For panel F, dots represent individual values, bars represent group mean + / - SEM. Unpaired two-way student t-tests were used to assess statistical significance between control and LB mice per age per sex in panel F. *=$p < 0.05$, **=$p < 0.01$, ***=$p < 0.001$.

The online version of this article includes the following figure supplement(s) for figure 6:

**Figure supplement 1.** Sex differences in mPFC to BLA projections in control reared mice.

PV positive cells, and a sex and age-specific increase in PL inputs to BLA. Further, LB altered levels of myelin basic protein at select developmental time points in the mPFC, effects that may contribute to altered development and functional readouts for behaviors that were not assessed here. Consistent with our results, studies in humans have found that childhood poverty decreases prefrontal activity in adulthood and reduces the ability of prefrontal cortex to suppress amygdala activity (*Kim et al., 2013*).

Given the developmental changes observed in LB reared mice, we sought to test if these effects may be related to changes in threat assessment and responding to fear associated stimuli. As the mPFC and BLA are critical for fear learning, mice at different developmental ages underwent Pavlovian tone-shock association learning (*Anglada-Figueroa and Quirk, 2005*; *Arruda-Carvalho and Clem, 2015*; *Etkin et al., 2011*; *Giustino and Maren, 2015*; *Sierra-Mercado et al., 2011*). Control reared mice showed the expected developmental trend of being able to condition to a tone-shock association at PND 21, but not at PND 16. Interestingly, LB reared animals were unable to express the conditioned memory 24 hr after conditioning at either PND 16 or PND 21, indicating a later developmental emergence of this behavior. At PND 21, LB mice were able to express the conditioned memory at a short delay (6 hr), indicating that deficits in freezing were not due to an inability to engage the freezing response. Furthermore, the freezing response re-emerged 7 days following conditioning, indicating that learning had occurred. Thus, LB may disrupt the developmental ability of animals to express, but not form, specific types of conditioned fear memories.

Previous research from our group and others have found effects of LB on depressive-like behavior in females (*Goodwill et al., 2019*), but failed to find significant differences in anxiety-like behavior in adult mice (*Goodwill et al., 2019*; *Manzano Nieves et al., 2019*; *Manzano-Nieves et al., 2018*; *Naninck et al., 2015*). Consistent with previous work, we did not find an effect of LB on general measures of anxiety. However, LB mice, when compared to Ctrls, had increased total time performing head dips in the elevated plus maze (EPM) at PND 21 and decreased number of entries in the light/dark box at PND 28 and 35, suggesting that LB rearing may be altering aspects of risk assessment, which were not directly tested in this study. Furthermore, we report no effects of LB rearing on auditory fear conditioning in young adults. This finding is in conflict with previous reports demonstrating a significant effect of LB rearing on auditory fear conditioning and expression (*Guadagno et al., 2018*). Differences between our fear conditioning results and *Guadagno et al., 2018* may be due to species differences (activity levels, learning differences, stress tolerance) or LB protocol (administration from PND 1–9 vs PND 4–11, cage changes versus no cage changes). It is possible that differences in LB paradigm may alter the frequency of abusive behaviors leading to differences in outcomes (*Gallo et al., 2019*). Detailed, meticulous studies comparing the timing of LB protocol administration and its effects on maternal behavior will be needed to understand how timing and duration of LB rearing impact auditory fear conditioning.

Previous research from our lab has demonstrated that LB can induce accelerated PV-positive cell differentiation in the hippocampus (*Bath et al., 2016*), but may also selectively decrease PV-positive cell counts in brain regions such as the orbitofrontal cortex in adult females (*Goodwill et al., 2018*). Consistent with reported effects in the hippocampus (*Bath et al., 2016*), we found that LB led to an earlier rise in PV-positive cell density in BLA, with PV-positive cell density being higher in LB reared mice compared to controls at PND 21. However, PV-positive cells are not increased across all brain regions or across all timepoints. Analysis of PV-positive cell density in rostral primary somatostatin cortex (S1) and rostral primary motor cortex (M1) revealed decreased cell counts in LB mice at PND 16 when compared to controls (*Figure 4—figure supplement 1*). The decreased PV-positive cell density in M1 and S1 were not persistent, with no effects being observed in S1 at any other age tested, and with increased PV-positive cell density being observed in M1 at PND 21 and 75. Together our data on PV-positive cell density across development and across brain regions revealed a mosaic effect of LB on PV-positive cell density. The mosaic effect may be the consequence of differences in the developmental timing of maturation of these brain regions, with seeding of PV-positive precursors occurring prior to the ELA manipulation in some areas. Alternatively, this may represent differing sensitivity of these regions to the stress signals (e.g. through differing regional expression of glucocorticoid (GR) and mineralocorticoid (MR) receptors, whose activation may have transcriptional effects impacting cellular maturation and plasticity). Future research testing potential mechanisms underlying region-specific sensitivity will be needed.

PV-positive neuronal activity in the BLA has been shown to be capable of modulating fear expression (*Davis et al., 2017*; *Wolff et al., 2014*). In addition, PV-positive cell density has previously been shown to be negatively correlated with innate threat responses in post-weanling rats (*Santiago et al., 2018*). Consistent with the previously described role of PV-positive cells in BLA, we found that mice reared under LB conditions had a decreased fear response, as indexed by lower freezing to a tone previously paired with a shock at PND 21. We posit that the precocious rise in PV positive cells in BLA may have blunted the ability to express, but not learn a fear association. In support of this interpretation of the data, optogenetic inhibition of PV-positive cells in BLA was sufficient to rescue the low freezing phenotype of PND 21 LB reared mice. However, other inhibitory neuronal populations within the BLA are also capable of modulating fear expression (*Krabbe et al., 2018*; *Lucas and Clem, 2018*; *Rovira-Esteban et al., 2019*; *Wolff et al., 2014*), thus, questions regarding the potential of other populations of neurons to be able to mask the effect of LB rearing remain to be explored. Although we did not observe differences between LB and control mice in somatostatin or VGLUT 2 cell densities at PND 21, this does not rule out the possibility of changes in physiological properties of these classes of cells. LB in rats has been shown to increase spine density and dendritic length of neurons and increase the excitability of BLA (*Guadagno et al., 2018*). Thus, additional studies will be needed to fully examine the role of the effects of LB rearing on diverse neuronal populations and on behaviors across development. The work included here significantly extends our knowledge of the role of PV positive cells in the development of fear expression, and possible mechanisms through which ELA can drive developmental changes in threat assessment and fear expression.

We show that LB increased PV-positive cell counts, and that this effect may have driven the decrease in fear expression at PND 21. However, it is also possible that LB-induced changes in mPFC to BLA anatomical connectivity may have contributed to the decreased freezing observed at PND 22. Previous studies in rats have shown that the mPFC can modulate auditory fear conditioning as early as PND 24, but not at PND 17 (*Kim et al., 2009*). Furthermore, it has been shown that stimulation of the IL subregion of the mPFC can decrease fear expression (*Do-Monte et al., 2015a*). Through the analysis of PL to BLA and IL to BLA projection densities, we found that LB females had increased PL to BLA estimated mean densities at PND 21 and 28. LB males did not significantly differ from Ctrl males at any age. Given that the deficits in fear expression at PND 22 are observed in both males and females, it is unlikely that differences in mPFC connectivity are driving the observed effects on cue associated fear learning. Additional experiments will be required to determine if increased PL to BLA anatomical connectivity in LB females is leading to increased functional connectivity, or the sex selective elevation in depressive-like behaviors we previously observed in females following LB rearing (*Goodwill et al., 2019*). Furthermore, future studies will be needed to examine the mechanisms and the consequences of the increased anatomical connectivity between these regions.

In sum, this work reveals complex effects of limited resource rearing, on the timing of neuronal and circuit maturation, and the developmental expression of fear learning. We found that resource restriction in the form of LB during early development can drive regional and cell selective effects on maturation, with profound implications for behavioral development. Specifically, we have found that LB increased PV-positive cell density in the BLA at PND 21, and transiently altered the ability of mice to express a threat-associated memory. The transient suppression of threat-associated fear behavior may increase risk taking (*Haushofer and Fehr, 2014*), increase the chances of incurring a secondary stressor, and mask emotional symptoms of early life trauma during the peri-weaning period, only to emerge as pathology in adolescence. Such observations may explain the latent period of neuropsychiatric symptom expression in children exposed to early life trauma (*Teicher et al., 2009*). Future studies assessing the contribution of altered development of these fear circuits to later symptom development will be needed.

# Materials and methods

**Key resources table**

| Reagent type (species) or resource | Designation | Source or reference | Identifiers | Additional information |
|---|---|---|---|---|
| Strain, strain background | $pvalb^{tm1(cre)Arbr}$ 'PV-Cre' | The Jackson Laboratory | RRID: IMSR_JAX:008069 | |
| Strain, strain background | $Gt(ROSA)26Sor^{tm39(CAG-hop/EYFP)/HZE}$ 'floxed Halo' | The Jackson Laboratory | RRID: IMSR_JAX:014539 | |
| Strain, strain background | $Sst^{tm2.1(cre)Zjh}$ 'somatostatin-Cre' | The Jackson Laboratory | RRID: IMSR_JAX:013044 | |
| Strain, strain background | $Slc17a6^{tm2(cre)Lowl}$ 'vGlut2-Cre' | The Jackson Laboratory | RRID: IMSR_JAX:016963 | |
| Strain, strain background | $Gt(ROSA)26Sor^{tm14(CAG-tdTomato)Hze}$ mice 'Ai14' | The Jackson Laboratory | RRID: IMSR_JAX:007908 | |
| Antibody | Anti-Parvalbumin (rabbit polyclonal) | Millipore Sigma | RRID: AB_838238 | (1:1,000) |
| Antibody | Anti-c-Fos (rabbit polyclonal) | Millipore Sigma | RRID: AB_2631318 | (1:20,000) |
| Antibody | Anti- β-Tubulin (mouse monoclonal) | Cell Signaling | RRID: AB_2715541 | (1:2,000) |
| Antibody | Anti-GAPDH (rabbit monoclonal) | Cell Signaling | RRID: AB_561053 | (1:2,000) |
| Antibody | Anti-Calretinin (mouse monoclonal) | Swant | RRID: AB_10000320 | (1:500) |
| Antibody | Anti-Calbindin (mouse monoclonal) | Swant | RRID: AB_10000347 | (1:500) |
| Antibody | Anti-Myelin Basic Protein (rabbit polyclonal) | Abcam | RRID: AB_1141521 | (1:1000) |
| Antibody | Anti-VGLUT1 (rabbit polyclonal) | Millipore Sigma | RRID: AB_2814811 | (1:1000) |
| Antibody | Anti-Mouse IgG (H+L) (donkey polyclonal) | Jackson Immuno Research | RRID: AB_2340770 | (1:2000) |
| Antibody | Anti-Rabbit IgG (H+L) (donkey polyclonal) | Jackson Immuno Research | RRID: AB_10015282 | (1:2000) |
| Chemical compound, drug | Cholera Toxin Subunit B (Recombinant), Alexa Fluor 594 Conjugate | Fisher Scientific | C22842 | 1.0 mg/ml |
| Software, algorithm | Noldus Ethovision XT | Noldus | RRID: SCR_000441 | Version 11 |
| Software, algorithm | SPSS | SPSS | RRID: SCR_002865 | Version 26 |

*Continued on next page*

*Continued*

| Reagent type (species) or resource | Designation | Source or reference | Identifiers | Additional information |
|---|---|---|---|---|
| Software, algorithm | GraphPad Prism | GraphPad | RRID: SCR_002798 | Version 8 |

## Subjects

Approximately 1091 C57BL/6N wildtype and 63 transgenic, male and female, mice were used in this study. Original breeding stock was ordered from Charles River Labs. All wild-type C57BL/6N mice were bred in house. For optogenetic experiments, *pvalb*$^{tm1(cre)Arbr}$'PV-Cre' (JAX#008069) and *Gt (ROSA)26Sor*$^{tm39(CAG-hop/EYFP)/HZE}$ 'floxed Halo' (JAX#014539) mouse lines were derived from a breeding stock acquired from Jackson laboratories. For optogenetic experiments, homozygous PV-Cre mice were bred with heterozygous floxed Halo mice resulting in two groups of offspring, *Het* PV-Cre/null floxed Halo (Light Controls) and *Het PV-Cre/Het* floxed Halo (PV Halo). For genetic labeling of select neuronal populations, homozygous *Sst*$^{tm2.1(cre)Zjh}$ '*somatostatin*-Cre' (JAX# 013044) or homozygous *Slc17a6*$^{tm2(cre)Lowl}$'*vGlut2*-Cre' (JAX#016963) mice were crossed with homozygous *Gt (ROSA)26Sor*$^{tm14(CAG-tdTomato)Hze}$ mice 'Ai14' (JAX#007908) mice to allow expression of the Ai14 reporter in a Cre-dependent manner. All animals were housed according to NIH guidelines and maintained on a 12 hr light:dark cycle. Lights were on from 7:30 am to 7:30 pm, with all experiments being conducted during the light period. Mice had free access to food and water throughout the study. All animal procedures were approved by the Brown University Institutional Animal Care and Use Committee and consistent with the National Institutes of Health Guide for the Care and Use of Laboratory Animals.

## Fragmented maternal care

LB was modeled through a resource restriction paradigm, in which dam and pups were placed in low bedding conditions with limited access to nesting material for 7 consecutive days (PND four through PND 11). This manipulation results in a fragmentation in maternal care (*Bath et al., 2016*; *Rice et al., 2008*). Four days after the birth of a litter (PND 4), the dam and pups were transferred from their standard home cage with cob bedding and a 4 × 4 cm cotton nestlet to an LB cage containing a wire mesh floor and a 2 × 4 cm cotton nestlet. The mice continued to have ad libitum access to food and water. Following 1 week (PND 11), pups and dams were returned to their standard housing with full bedding and nesting material. Standard reared mice (designated as Controls—Ctrl) were left undisturbed in a standard home cage until weaning. All pups were weaned and sex segregated at PND 21, with the exception of mice tested at PND 21, which were weaned following the completion of fear conditioning experiments at PND 22.

## Mouse body and brain weight

To analyze mouse body and brain weights mice were deeply anesthetized with pentobarbital (Beuthanasia 150 mg/kg IP). Mice were first weighed to obtain the full body weight, then the brains were quickly removed. To ensure that brain collection was complete and carried out in an identical manner between groups, the brain stem was cut at the level of the occipital bone and the premaxila and nasal bones were crushed at the rostral most level of the eyes. This allowed us to remove the whole brain with the cerebellum and the intact olfactory bulbs. Brains were then weighed. One mouse was sacrificed at a time to ensure minimum protein degradation as brains were subsequently flash frozen prior to protein extraction for western blot analysis.

## Fear conditioning

Fear conditioning was carried out in Med Associates (St. Albans City, VT) operant chambers. On days 1 and 2, mice were habituated to two distinct (differing in color, texture, and smell) chambers. Habituation trials lasted 5 min per chamber and were counterbalanced. On day 3, mice received tone-shock associative learning in the fear conditioning chamber. During fear conditioning, mice were presented with six tones (30 s, 4 KHz, 75 dB) with each tone co-terminating with a 1 s footshock (0.57 mA). Tone shock pairings were separated by an inter-trial interval of 1.5 min. Testing for fear expression occurred on day 4 of the testing protocol, unless otherwise stated. Fear expression

testing consisted of exposing mice to two tones in the control chamber (habituated chamber where no fear conditioning occurred). Different cohorts of animals, across multiple litters were used to test mice at the different developmental time points. Freezing behavior was scored automatically by the activity tracker module in Noldus Ethovision XT 11.0 and verified from video by observers blind to treatment and condition but not to age (as age could be inferred based upon differences in mouse size).

## Light/dark box

Mice were tested in a Light/dark box that was built in house. To begin a trial, mice were placed in the dark side of the box which was connected to a light side by a small opening (~6×6 cm). To increase the brightness of the light side (~139 Lux), a lamp, pointing toward the light side, was mounted on the lid of the dark side. The dimensions of the dark and light side of the chamber were the same and measured (height = 23 cm, width = 22 cm, length = 26 cm). A trial lasted a total of 10 min. Activity of the mouse during a trial was recorded and analyzed using Ethovision XT 11.0 software, with latency to first exit being hand scored by an observer blind to sex, age, and condition.

## Immunohistochemistry

To assess the relative density of PV-positive cells and c-Fos expressing cells, immunohistochemistry was performed on control and LB mice on the days stated in each experiment. Briefly, mice were deeply anesthetized with pentobarbital, transcardially perfused with buffered saline followed by 4% paraformaldehyde, and processed for immunohistochemistry as previously described (*Bath et al., 2016*). For PV-positive cell labeling, a rabbit anti-parvalbumin antibody (1:1,000; Millipore) was used. For c-Fos labeling a rabbit anti c-Fos (1:20,000; Millipore) was used. Brain sections (40 μm) were mounted on charged glass slides, counter stained using a Hema three staining set (Fisher Scientific Company), dehydrated, and coverslipped for imaging. For prelimbic, infralimbic, rostral primary motor cortex, and rostral primary somatosensory cortex, brain sections from AP 1.94 to 1.54 were analyzed. For the amygdala, brain sections from AP −1.22 to −2.06 were analyzed.

## Western blot

Male mice were sacrificed, brains were quickly dissected, weighed, and flash frozen on dry ice. The medial prefrontal cortex (mPFC), and the basolateral amygdala (BLA) were dissected and stored at −80°C until processing.

Tissues were homogenized in RIPA buffer (with 1% protease and phosphatase inhibitor cocktail, Fisher Scientific) and supernatant was collected following centrifuging at 14,000 rpm at 4°C for 10 min. Protein concentration was determined with a bicinchoninic acid (BCA) kit (Thermo Scientific, Waltham, MA). Protein lysates were each diluted to 1.0 mg/mL with RIPA buffer, heated at 90°C for 10 min, and proteins separated by gel electrophoresis on a 12% sodium dodecyl sulfate–polyacrylamide gel electrophoresis (SDS-PAGE) gel, and transferred to polyvinylidene difluoride (PVDF) membrane. For the remaining portion of the western blot protocol, transfer membranes were kept at 4°C.

Membranes were blocked for 1 hr in 5% non-fat milk in Tris-buffered saline Tween-20 (TBST, containing 10 mM Tris, 150 mM NaCl, and 0.1% Tween-20, pH 7.6), followed by incubation with primary antibodies diluted in 5% non-fat milk/0.5% bovine serum albumin in TBST at 4°C overnight. Membranes were washed with TBST three times (15 min per wash) and incubated with secondary antibody in 5% non-fat milk/0.5% bovine serum albumin in TBST for 1 hr. Membranes were then washed with TBST three times (15 min per wash), then visualized with Amersham ECL Western Blotting Detection Reagent (RPN2106, GE Life Sciences) using a C600 Azure Biosystems imaging system (Dublin, CA). Densitometry analysis was conducted with a gel imaging module of NIH ImageJ software.

### Primary antibodies used for western blotting

Mouse anti-β-Tubulin (1:2000, Cell Signaling Technology), rabbit anti-GAPDH (1:2000, Cell Signaling Technology), mouse anti-Calretinin (1:500, Swant), mouse anti-Calbindin (1:500, Swant) rabbit anti-Myelin Basic Protein (1:1000, Abcam), rabbit anti-VGLUT1 (1:1000, Millipore). Secondary antibodies

used for this study were: HRP conjugated donkey anti-mouse (1:2000, Jackson ImmunoResearch) and donkey anti-rabbit (1:2000, Jackson ImmunoResearch).

## Cholera toxin B injections

Alexa 594 conjugated cholera toxin B (CTB) (Fisher Scientific) was used to retrogradely label the projections from PL and IL to BLA, in male and female mice, at postnatal ages PND 16, 21, 28, and 35. CTB (1.0 mg/mL) was injected (0.15 ul) into the left BLA 1 day prior to the time-point of interest (e.g. PND 15 for PND 16). In order to inject the CTB into BLA across development separate coordinates were used for each age. Developmentally appropriate coordinates were empirically derived from pilot surgeries. Coordinates used for the CTB injections were as follows: (PND 15: DV = −5.075, ML = −3.05, AP = −1.1; PND 20: DV = −5.1, ML = −3.1, AP = −1.15; PND 34: DV = −5.2, ML = −3.15, AP = −1.2). Mice were perfused 48 hr post-injection. The brain was dissected, sectioned (40 μm), mounted, counter-stained with DAPI (Fisher Scientific) and visualized using a fluorescent microscope. The density of CTB positive cells in the PL and IL was measured.

## Optogenetic surgery and inhibition of PV+ cells

Female mice homozygous for Cre under the control of a parvalbumin driver (JAX#008069- $Pvalb^{tm1(cre)Arbr}$) were crossed with a male heterozygous floxed Halo (JAX#014539- $Gt(ROSA)26Sor^{tm39(CAG-hop/EYFP)Hze}$). The cross resulted in the selective expression of halorhodopsin in PV+ cells (Het PV-Cre, Het Halo 'PV Halo mice') and mice from the same litter that were Cre positive, but lacked the optogenetic channel (Het PV-Cre, null floxed Halo 'Light control'). Mice were bilaterally implanted with an in-house made ceramic optic fiber (Ø200 μm Core, 0.50 NA; Thorlabs, Newtown, NJ) at PND 15 above BLA (Placements: DV = −5.1, ML = = + / - 3.1, AP = −1.15). For surgeries, mice were anesthetized with isoflurane gas anesthesia (2.0%–2.5% in 1 l/min oxygen) and secured to the stereotaxic apparatus. The scalp was shaved and cleaned, Buprenex (0.1 mg/kg, as an analgesic) was administered intraperitoneally and lidocaine was applied. The skin above the skull was removed to expose the skull. A ~1 mm–diameter craniotomy was drilled above BLA. The right side implant was lowered and temporarily secured with Metabond (Parkell Inc, Brentwood, NY) while the left side fiber optic was placed. Following the implantation of the second optic fiber, a thin layer of metabond was placed above the skull to adhere the dental acrylic to the skull. After the dental adhesive solidified, isoflurane administration was stopped, and mice were allowed to wake.

Mice began fear conditioning protocol at PND 19 (as described above), with optogenetic inhibition occurring at PND 21 (*Figure 5*). During conditioning, PV+ cells in the BLA were photo-inhibited with constant light (620 nm Plexbright LED, Plexon, Dallas, TX), using an LED driver (Plexon, Dallas, TX) during the 30 s of the tone (including the 1 s foot-shock). The light power delivered, as measured through the optic fiber pre-implant, ranged from 1.5 to 2 mW per side. Following the fear conditioning protocol, a random subset of the mice were tested for locomotion in an open field under conditions of light stimulation.

## Microscopy

Neurolucida software was used to analyze immunohistochemical data. Either a light (for DAB staining) or epi-fluorescent microscope (for fluorescence) was used when appropriate. For quantification of neuronal cell density, brain regions were traced at 4x magnification and borders were defined as shown in Paxinos and Franklin mouse brain atlas. Immunoreactive or fluorescent positive neurons within each region were identified by an observer blind to condition and treatment (10x). All region contours with identified cells were saved and the number of cells and area within each contour was assessed using StereoInvestigator. For each brain region, 3–4 sections per brain were averaged to obtain a mean density.

## Statistical analyses

A two-tailed student's *t-test* was used to compare between two groups. When more than two groups were assessed the appropriate ANOVA was performed as stated in the figure legends. All ANOVA tests were followed by Sidak's multiple comparison test, assessing the effects of treatment at each given age and/or assessing developmental differences within each treatment.

Two separate analyses of CTB retrograde injection data are presented in *Figure 6*. For *Figure 6D–E*, the density of CTB-positive cells in PL and IL (obtained as described in the microscopy section above), was analyzed through the use of an ANCOVA analysis. Two covariates were used in the ANCOVA analysis: 1) the area of the BLA at the site of the injection and 2) the area of the injected CTB at the site of the injection. The site of the injection was defined as the brain section where the needle track was at the ventral most point. The area of the injected CTB was used as a correction for injected volume, while the area of BLA at the injected site was used as a correction for rostral - caudal placement, as the BLA increases in area as you move caudally. Using the two covariates, the ANCOVA analysis performed per age per sex per region, returned an estimated mean value, the 95% confidence intervals, and the pairwise comparison statistics presented in *Figure 6D–E*. A secondary analysis, projection index, correcting for efficiency of labeling is presented in *Figure 6F*. For this analysis, the density of CTB labeling (obtained as described in the microscopy section above) in IL was subtracted from PL (PL-IL) and divided by the total number of cells labeled (PL+IL). The projection index [(PL-IL)/(PL+IL)] thus accounts for differences in labeling efficiencies between mice. This data is presented as mean + / - SEM, with statistical significance being determined by a two-tailed student t-test conducted per age, per sex, as detailed in the figure legend.

With the exception of data in *Figure 6D–E*, Statistical analysis was performed using Prism Graphpad statistical and graphing software. *Figure 6D–E* was analyzed with SPSS statistical software. All data were graphed using Prism Graphpad statistical and graphing software. For all statistical analyses, statistical significance was defined as $p < 0.05$.

## Methods relevant to supplementary figures
### Open field test
To test for differences in locomotor activity and anxiety-like behavior, mice were placed in an open field arena as previously described (*Goodwill et al., 2019*). Distance moved and the time spent in the center of the arena were recorded during a 7 (*Figure 3—figure supplement 3*) or 5 (*Figure 5—figure supplement 2*) minute test using the Ethovision video-tracking system. The arena was digitally divided into two zones (center and periphery), as previously described (*Goodwill et al., 2019*). Decreased time in the center was used as an indicator of anxiety-like behavior.

### Elevated plus maze
To assess anxiety-like behavior at PND 21 mice were placed in an elevated plus maze as previously described (*Manzano Nieves et al., 2019*; *Manzano-Nieves et al., 2018*). The EPM consisted of two open (unprotected) and two closed (protected) arms. Greater time in the closed (protected) arms is defined as higher anxiety-like behavior. Mice were allowed 7 min to explore the maze. The time spent in the protected vs. unprotected arms was assessed. Videos were recorded, and behavior was tracked using Noldus Ethovision XT 10.0 software. All trials were conducted under low-light conditions (~109 Lux). Time spent and distance walked in the open and closed arms of the EPM were assessed using the mouse tracking module. The amount of time, and number of instances, that the animals spent with their head below the surface of the open arms (Head Dips) was manually scored by an independent observer blind to the sex and condition of the mice.

### Shock sensitivity assay
To assess the minimum foot-shock intensity required to elicit a behavioral response (visible flinch or audible vocalization), PND 21 mice were placed in an operant conditioning chamber (Med associates, Fairfax, VT). Mice were exposed to a series of foot shocks, beginning at 0.06 mA and increasing at 0.02 mA intervals (*Manzano-Nieves et al., 2018*). Each shock intensity was presented three times. The amplitude of the foot-shock at which a given mouse first flinched, and/or audibly vocalized to 2 out of 3 foot shocks at a given intensity was recorded by two independent observers blind to condition to insure agreement on these measures. Flinching was defined as the mouse moving its body reflexively downward, making its body smaller, directly following a foot-shock. Vocalization was defined as the emittance of an audible sound.

## Acknowledgements

This work was supported by funding by the National Institutes of Health, from the NIMH (MH115914- KGB; MH115049- KGB) and NINDS (NS105219- GMN). The authors thank Angelica Johnson for providing the hand scored data for the EPM, as well as Hyeyoung Shin and Roberto Andres Aponte Rivera for providing valuable feedback on the project. We thank the Barry Connors lab and the Christopher Moore lab at Brown University for facilitating access to the transgenic mice, and the Rebecca Burwell lab for permitting us the use of their lab's microscope.

## Additional information

### Funding

| Funder | Grant reference number | Author |
| --- | --- | --- |
| National Institutes of Health | MH115914 | Kevin G Bath |
| National Institutes of Health | MH115049 | Kevin G Bath |
| National Institutes of Health | NS105219 | Gabriela Manzano Nieves |

The funders had no role in study design, data collection and interpretation, or the decision to submit the work for publication.

### Author contributions

Gabriela Manzano Nieves, Conceptualization, Data curation, Formal analysis, Methodology, Writing - original draft; Marilyn Bravo, Supervision, Investigation, Methodology, Writing - review and editing; Saba Baskoylu, Investigation, Methodology, Writing - review and editing; Kevin G Bath, Conceptualization, Resources, Supervision, Funding acquisition, Investigation, Methodology, Project administration, Writing - review and editing

### Author ORCIDs

Gabriela Manzano Nieves https://orcid.org/0000-0001-6989-4872
Kevin G Bath https://orcid.org/0000-0003-2229-177X

### Ethics

Animal experimentation: This study was performed in strict accordance with the recommendations in the Guide for the Care and Use of Laboratory Animals of the National Institutes of Health. All of the animals were handled according to approved institutional animal care and use committee (IACUC) protocols (#19-10-0003) of Brown University.

### Decision letter and Author response

Decision letter https://doi.org/10.7554/eLife.55263.sa1
Author response https://doi.org/10.7554/eLife.55263.sa2

## Additional files

### Supplementary files

• Transparent reporting form

### Data availability

Data has been deposited in the Brown Digital Repository with the following https://doi.org/10.26300/9krc-h052.

The following dataset was generated:

| Author(s) | Year | Dataset title | Dataset URL | Database and Identifier |
|---|---|---|---|---|
| Bath KG, Nieves GM, Bravo M, Baskoylu S | 2020 | Data from Early life adversity decreases pre-adolescent fear expression by accelerating amygdala PV cell development | https://doi.org/10.26300/9krc-h052 | Brown Digital Repository, 10.26300/9krc-h052 |

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
