## [Decision Letter]

Thank you for submitting your article "Early life adversity decreases fear expression in pre-adolescence by accelerating amygdalar parvalbumin cell development" for consideration by *eLife*. Your article has been reviewed by three peer reviewers, one of whom is a member of our Board of Reviewing Editors, and the evaluation has been overseen by Kate Wassum as the Senior Editor. The following individual involved in review of your submission has agreed to reveal their identity: Claire-Dominique Walker (Reviewer #3).

The reviewers have discussed the reviews with one another and the Reviewing Editor has drafted this decision to help you prepare a revised submission.

This manuscript examined the impact of early adversity, specifically that caused by resource insecurity, here modeled by limited bedding, on physical and neural development and the trajectory of emotional learning. The experiments are conducted over an age period adequately spanning the late neonatal, early juvenile, adolescent and adult age, in line with the period of postnatal maturation of anatomical connections between basolateral amygdala and prefrontal cortex. This kind of time course work is essential because snapshots of behavior or brain function at single time points either completely miss effects or lead to erroneous conclusions as to the direction of effect. Limited bedding impaired somatic and neural development at a gross level, but resulted in a transient suppression of fear memory recall, which occurs coincident with an elevation in parvalbumin cell density in the basolateral amygdala. This impairment in fear recall was rescued by optogenetic inactivation of basolateral amygdala parvalbumin neurons during fear learning. Thus limited bedding-induced accelerated development of basolateral amygdala parvalbumin neurons seems not influence fear memory consolidation, but does impair recall of the memory the following day. Overall, these studies are carefully run, with internal replication built in and thoughtful data analysis.

Overall, all of the reviewers were positive about the research in the manuscript, and there was only one essential revision issue that was brought up by all reviewers which requires attention, and this is the lack of an appropriate no-LB control optogenetic study on memory recall to determine if the effects of inhibition of PV+ neurons in the BLA have an influence in control mice. Given the nature of this study, it was felt that the addition of this control group would be achievable within the designated timeframe, and so all reviewers agreed this was an essential revision.

In addition to this revision there are several other more minor issues which the reviewers felt required attention and they are listed below. Please address all of these issues in your response letter.

Reviewer comments to address:

1) It is not clear if there has been work done on the fidelity of the mouse Cre line being used demonstrating that halorhodopsin expressing cells are indeed PV+ GABA interneurons. If so, could the authors mention a citation for this, and if not, it would be probably good for the authors to at least show colocalization for Cre expression with vGAT and PV.

2) Why was constant light used to drive halorhodopsin inhibition rather than light pulses? There is evidence that constant light overheats tissue and leads to inadvertent transcriptional activity if tissue is not damaged. For prolonged inhibition of activity, a chemogenetic approach may be more desirable. New work is not necessary (suggested for future) but justification of constant light in methods is needed.

3) It makes sense that releasing PV inhibition in the BLA could enhance freezing behavior. While the authors do not find an effect of ELA on other inhibitory neurons, a valuable control would still be testing whether silencing of any other inhibitory cell type in the BLA produces a similar enhancement of freezing. Given the significant extra work to provide evidence of specificity, authors may instead choose to note possibility/limitations in the text.

4) Why was interneuron maturation assessed by IHC for PV-expressing neurons but by Western for other classes of cells?

5) Attempts to control for CTB injection differences by covarying for injection area are to be commended. It would also be useful to calculate a simple ratio of projection density / injection area to see if significant differences remain.

6) Please remove assertions that the limited bedding mouse paradigm models poverty and/or refugee experience, which is significantly more complex in humans and involves layers of society and not just “disorganized” care. Bedding is only one resource that is restricted for the mice, whereas human poverty impacts nutritional quality, social stability, educational opportunities, etc.

7) Additional discussion of why/how the LB paradigm could accelerate PV+ maturation in some brain regions and delay it in others (encompassing several papers from this lab) is warranted.

8) Why would LB from 4-11 only affect density measures at P35, and only alter ratio at PN21 (Figure 5)? It is not clear how the authors interpret these delayed effects and differences in timing.

9) Reduced body weight gain has been demonstrated in virtually every study using the LB paradigm in both mice and rats. Ratio of brain weight over body weight nicely shows that there is possibly an element of brain sparing to this effect. In Figure 1, data for PND16 should be drawn on a different scale as it is difficult to evaluate the group difference at that age. Obvious and significant differences are reported for older ages. Figure 1—figure supplement 1, the color code for males is confusing and not well contrasted. Brain/body weight is certainly one index of somatic development, but it would have been nice to complement with anogenital distance in both sexes. Do the authors have data pertaining to this variable between controls and LB mice?

10) How exact is the assumption that PV+ cell number is associated with changes in cell differentiation? The authors indicate that LB accelerates the differentiation of PV in the BLA, however, differences could also be due to the level of immunostained signal in the BLA at young ages. Replace the term "differentiation" by increased PV-positive cell number.

11) In Figure 3, the standard errors for PV density are very large for PND21 in the BLA. There is a clear overlap of the standard errors suggesting that the Ctl and LB groups are not significantly different. In this case, the authors should correct their statistical analyses to perform a 2 way ANOVA (age x treatment) followed by post-hoc tests instead of a series of t-tests. This is a critical point as it is one of the main results of the manuscript.

12) The experiments with optogenetic inhibition of PV+ cells in the BLA are very elegant and carefully designed. In Figure 4C, placement of the optical fiber suggests that the LA rather than the BLA was inhibited. Disinhibition of the LA will likely lead to BLA activation and a similar increase in %freezing, but this should be discussed. Please provide the bregma level for the fiber optic placement. Provide a better photomicrograph of mPFC GFP staining. Even with higher magnification, there are no cells that can be detected.

13) ".…the impact of LB on mPFC and BLA connectivity remained unknown." This is inaccurate as other studies have examined connectivity between these two structures after ELA. Please mention and reference these important imaging (DTI and rsfMRI) studies: Bolton et al., 2018, Yan et al., 2010 and Guadagno et al., 2018. The authors could precise: "anatomical connectivity". Throughout the manuscript "anatomical connectivity" should be used. It is unclear how the CTB injection size was determined? In fact, it should be a volume of injection since several sections likely contained CTB. How projection density was normalized should be explained in greater details in methods.

14) Please provide high resolution photomicrographs of the injection site in the BLA as well as the projection areas in mPFC. What is the background for this type of labeling? Please indicate whether you found layer differences in the density of labeling in the mPFC IL and PL.

15) The authors provide figures in the manuscript where data from both sexes are pooled, yet in the supplementary figures, there are significant effects of sex and age on the %freezing during conditioning. I wonder whether these data should not be presented in the manuscript as well as complement to Figure 2. Interestingly also, Figure 2—figure supplement 1 fails to find significant differences in %freezing behavior between Ctl and LB adult mice (both male and female). This is surprising given the assumption that ELA leads to phenotypes of increased anxiety and fear responsiveness as demonstrated by several other groups in similar conditions. This should be discussed.

---

## [Author Response]

Reviewer comments to address:1) It is not clear if there has been work done on the fidelity of the mouse Cre line being used demonstrating that halorhodopsin expressing cells are indeed PV+ GABA interneurons. If so, could the authors mention a citation for this, and if not, it would be probably good for the authors to at least show colocalization for Cre expression with vGAT and PV.

We thank the reviewer for raising this point. The group that generated this PV-Cre line demonstrated that more than 90% of PV+ cells are labeled in this mouse line (Hippenmeyer et al., 2005 PLoS Biol). Further, as part of a collaboration for a separate project with Dr. Barry Connors (Goodwill et al., 2019), we have characterized the intrinsic properties of PV Cre x NpHR positive cells in the orbitofrontal cortex (OFC) of slices obtained from this line of mice (unpublished data). Based on characterizing the intrinsic property of PV-Cre cells in the OFC, we are confident that the clear majority of the cells (all cells patched to date) exhibit all of the properties of GABAergic PV+ cells. Similar work has been carried out using this same PV-Cre mouse line in combination with AVV-NpHR-EGFP cell infection, to show selective silencing of PV+ cell populations (Royer et al., 2012 Nat Neuro).

2) Why was constant light used to drive halorhodopsin inhibition rather than light pulses? There is evidence that constant light overheats tissue and leads to inadvertent transcriptional activity if tissue is not damaged. For prolonged inhibition of activity, a chemogenetic approach may be more desirable. New work is not necessary (suggested for future) but justification of constant light in methods is needed.

We agree with the reviewer and had a similar concern prior to undertaking these studies, which motivated our specific approach. We previously carried out in vitro optogenetic inhibition of parvalbumin positive cells using NpHR in slices derived from this same mouse line (aforementioned collaboration with Dr. Connors, also see publication- Goodwill et al., 2019). Recordings were carried out at three different current intensities in the same cell from a PV Cre x NpHR mouse. Patched cells had all of the properties of fast spiking PV+ cells under both patch and current stimulation conditions. We found that spiking of these cells could be immediately silenced upon light stimulation, that this silencing was maintained through a 20 minute protocol, and that normative spiking resumed upon light removal, with no rebound inhibition, no enhanced spiking following release, and no obvious effects on the intrinsic properties of these cells. We would like to also note that in our protocol, we employed the use of light generated from LED drivers, which provide lower power, but effective modulation of NpHR. Use of lower power light sources may eliminate potential heating or transcriptional effects that may be observed with continuous higher power light stimulation when using lasers as the light source. Despite this, we have now included a caveat about the use of constant light and possible benefits of using other approaches.

3) It makes sense that releasing PV inhibition in the BLA could enhance freezing behavior. While the authors do not find an effect of ELA on other inhibitory neurons, a valuable control would still be testing whether silencing of any other inhibitory cell type in the BLA produces a similar enhancement of freezing. Given the significant extra work to provide evidence of specificity, authors may instead choose to note possibility/limitations in the text.

We thank the reviewers for this comment and agree that there may be potential effects of silencing of other populations of cells on freezing behavior. For example, in adult animals, manipulation of CCK cells alters engagement of freezing behavior (Rovira Esteban et al., 2019). We agree that the manipulation of activity of a number of other cellular populations may in fact drive changes in freezing behavior (augmentation or inhibition). Our studies, and the choice of inhibition of PV positive cells in BLA, were focused on the observation of altered PV positive cell density at early developmental timepoints, and the attempt to provide proof of concept that they could be influencing freezing levels by releasing the believed net effect of an increased PV influence on regional activity at this age. Our observations support the argument that decreasing the influence of PV at this age can restore freezing behavior, and in the anticipated direction. Experiments manipulating other classes of cells could have interesting and important effects and add to our understanding of the effects of activity of these cells in regional activity and will be important studies to carry out in the future. However, we believed that such studies were outside of the scope of the current manuscript, as we had observed limited effect of LB rearing on gross metrics of cellular maturation for other classes of cells (e.g. SST and vGLut). Experiments silencing SST+ cells may also support changes in freezing behavior, which would be interesting, but may merely represent another mechanism of altering behavioral output, but one that was not likely impacted by LBN rearing. We have added text indicating this caveat and the need to look at the influence of manipulating additional classes of cells on this behavior over development.

4) Why was interneuron maturation assessed by IHC for PV-expressing neurons but by Western for other classes of cells?

We thank the reviewers for bringing up this important point. We had planned to use multiple methodologies for quantification of cell density over development in the varied classes of cells. However, for IHC approaches, the PV antibody was the most robust marker used across development. Work with other antibodies provided exceedingly poor staining in tissue at the early developmental ages that were being assessed (possibly due to developmental changes in antigen availability). Variability in the quality of IHC hampered our ability to conduct such developmental assays for a broader range of markers. Given these limitations, we moved to using alternate methods to verify results for glutamatergic neurons and somatostatin positive neurons. We used additional mice that genetically express Cre recombinase under either the somatostatin (SST) or glutamatergic (Vglut2) promoter. We crossed each of these lines of mice with an Ai14 (Td-tomato reporter mouse). Using this method, we verified a subset of the findings showing no effect of LB rearing on SST+ or vGlut2^+^ cell density in BLA at P21. This is now included in the supplemental data.

5) Attempts to control for CTB injection differences by covarying for injection area are to be commended. It would also be useful to calculate a simple ratio of projection density / injection area to see if significant differences remain.

Thank you for this suggestion. After running the requested analysis we found that increased PL to BLA projections at PND 21 (p = 0.000757) remained significant while differences at PND 28 were no longer significant (p = 0.099). This analysis strategy did not impact statistical significance for other comparisons.

6) Please remove assertions that the limited bedding mouse paradigm models poverty and/or refugee experience, which is significantly more complex in humans and involves layers of society and not just “disorganized” care. Bedding is only one resource that is restricted for the mice, whereas human poverty impacts nutritional quality, social stability, educational opportunities, etc.

We thank the reviewer for this point and agree that the human condition is much more complex than what is modeled with the limited bedding intervention in mice. We have deleted the requested assertion and have revised the manuscript to reflect that we are observing effects of loss of maternal resources on pup care and development, and do not overreach with our description of this mimicking poverty/refugee experiences.

7) Additional discussion of why/how the LB paradigm could accelerate PV+ maturation in some brain regions and delay it in others (encompassing several papers from this lab) is warranted.

We thank the reviewer for this point and have added brief discussion of this in a recent review paper (Bath 2020, TINS). We now include additional supplemental data showing that LB leads to significant delays in the developmental increase in PV positive cell density in the primary motor and primary somatosensory cortex, supporting those assertions. We have also added a brief discussion of why regionally specific effects may be occurring.

8) Why would LB from 4-11 only affect density measures at P35, and only alter ratio at PN21 (Figure 5)? It is not clear how the authors interpret these delayed effects and differences in timing.

This is an excellent question, which we are currently working to understand. Based upon our prior work in hippocampus and other brain regions, we know that LB from P4-11 may be impacting the migration (less likely) or timing of maturation and differentiation in at least some classes of cells (we focused on PV positive cells). In that prior work, we have found in control animals that PV expression begins to elevate in these regions well after the LB manipulation (P16- Bath et al., 2016) and not reaching peak levels until ~P28. Further, this inhibition likely plays a role in the maturation of the local circuit, functioning in the stabilization and pruning of later developing projections from other brain regions. Thus, LB rearing may have cascading downstream effects on additional maturational processes, which may not manifest until later timepoints. Further, some of these effects may persist, or resolve due to ongoing plasticity in the system, or utilizing alternate strategies and computations to drive shifts in behavior.

9) Reduced body weight gain has been demonstrated in virtually every study using the LB paradigm in both mice and rats. Ratio of brain weight over body weight nicely shows that there is possibly an element of brain sparing to this effect. In Figure 1, data for PND16 should be drawn on a different scale as it is difficult to evaluate the group difference at that age. Obvious and significant differences are reported for older ages. Figure 1—figure supplement 1, the color code for males is confusing and not well contrasted. Brain/body weight is certainly one index of somatic development, but it would have been nice to complement with anogenital distance in both sexes. Do the authors have data pertaining to this variable between controls and LB mice?

We have now included an inset in panels A and B of Figure 1 that provides a zoomed in version of PND 16 data. We have changed the colors in the Figure 1—figure supplement 1 to enhance contrast. We agree that anogenital distance would be a useful complementary measure of somatic development. However, the anogenital distance for Ctrl and LB mice was not recorded as part of this study and we are unable to obtain this data at this time. However, we will strive to include it as part of our developmental measures for future studies.

10) How exact is the assumption that PV+ cell number is associated with changes in cell differentiation? The authors indicate that LB accelerates the differentiation of PV in the BLA, however, differences could also be due to the level of immunostained signal in the BLA at young ages. Replace the term "differentiation" by increased PV-positive cell number.

We have amended the manuscript to reflect the reviewer’s comment. The increase in PV positive cells was confirmed with a reporter mouse line that would be less sensitive to fluctuating PV levels from immunostaining. The term “differentiation” was somewhat loosely used to mean the developmental change in the expression of the PV protein, which we interpreted as the differentiation of this class of cells to express a PV phenotype, not necessarily differentiation from precursor to mature phenotype. To address the reviewer concern, we have now replaced the term differentiation with “increased PV-positive cell density”.

11) In Figure 3, the standard errors for PV density are very large for PND21 in the BLA. There is a clear overlap of the standard errors suggesting that the Ctl and LB groups are not significantly different. In this case, the authors should correct their statistical analyses to perform a 2 way ANOVA (age x treatment) followed by post-hoc tests instead of a series of t-tests. This is a critical point as it is one of the main results of the manuscript.

We thank the reviewers for bringing this to our attention. There was an issue in the rendering of the figure due to last minute changes in the manuscript between the editorial review and the scientific review process, where the graph was accidentally changed to depict standard deviation instead of standard error of the mean. We have now corrected this issue to show the mean and SEM for the data presented. We agree with the reviewer that a 2-way ANOVA would have been a better statistical approach. However, due to logistical issues associated with the number of animals and sections prepared, immunostaining was performed separately for each age. This meant that we could only ensure that the various factors associated with differences in quality of labeling (e.g. incubation durations, separate preparations of primary antibody incubation solution, washes, separate preparation of secondary incubation reagents, and development reagents and incubations) were matched within an age group but not across ages. Because of the way the data was generated we did not feel that it was appropriate to include analysis across ages. The same is true for the western blots.

12) The experiments with optogenetic inhibition of PV+ cells in the BLA are very elegant and carefully designed. In Figure 4C, placement of the optical fiber suggests that the LA rather than the BLA was inhibited. Disinhibition of the LA will likely lead to BLA activation and a similar increase in %freezing, but this should be discussed. Please provide the bregma level for the fiber optic placement. Provide a better photomicrograph of mPFC GFP staining. Even with higher magnification, there are no cells that can be detected.

In accordance with previous work done by members of this research team (Goodwill et al., 2018, Do Monte, 2015) we aimed to place optical fibers on the surface of the brain region to be optically inhibited. However, we do recognize that in this study, the placement of the fibers has the potential to also influence cellular activity in LA. We have now added this caveat to the discussion of our results.

The targeted placement of the optical fibers were as follows: DV = -5.1, ML = = +/- 3.1, AP = -1.15 and is included in the Materials and methods section.

We agree that it was hard to identify the PV+ cells within mPFC and now provide a zoomed in image as part of Figure 5.

13) ".…the impact of LB on mPFC and BLA connectivity remained unknown." This is inaccurate as other studies have examined connectivity between these two structures after ELA. Please mention and reference these important imaging (DTI and rsfMRI) studies: Bolton et al., 2018, Yan et al., 2010 and Guadagno et al., 2018. The authors could precise: "anatomical connectivity". Throughout the manuscript "anatomical connectivity" should be used. It is unclear how the CTB injection size was determined? In fact, it should be a volume of injection since several sections likely contained CTB. How projection density was normalized should be explained in greater details in methods.

We have now provided a more precise description of anatomical connectivity throughout the manuscript. We apologize for the lack of discussion of these additional studies and have now included the recommended citation including additional citation of the human literature. We have also included how the data was normalized in greater detail with the Materials and methods section.

14) Please provide high resolution photomicrographs of the injection site in the BLA as well as the projection areas in mPFC. What is the background for this type of labeling? Please indicate whether you found layer differences in the density of labeling in the mPFC IL and PL.

An example image of the CTB injections into BLA and the cell labeling in mPFC has now been included in Figure 6. Unfortunately, due to limitations at the time of data collection, no layer information was recorded for this dataset.

15) The authors provide figures in the manuscript where data from both sexes are pooled, yet in the supplementary figures, there are significant effects of sex and age on the %freezing during conditioning. I wonder whether these data should not be presented in the manuscript as well as complement to Figure 2. Interestingly also, Figure 2—figure supplement 1 fails to find significant differences in %freezing behavior between Ctl and LB adult mice (both male and female). This is surprising given the assumption that ELA leads to phenotypes of increased anxiety and fear responsiveness as demonstrated by several other groups in similar conditions. This should be discussed.

We thank the reviewer for this request, as we had collected data in both male and female mice for the bulk of our measures, because we are very interested in potential sex differences (Bath 2020- TINS). In prior manuscripts, reviewers have had difficulty with the either the discussion of the complexity associated with sex differences or the additional length of results and discussion when separating out sex specific data, and thus we tried to keep the focus on the main effect (suppressed freezing at P21). We now have happily amended Figure 2 to show the sex specific effects and separated out male and female data in other figures. We have also included a discussion addressing the lack of significant increases in fear expression and anxiety-like behaviors within our LB reared mice. Previous research from our group (Goodwill et al., 2019, Manzano Nieves et al., 2018, Manzano Nieves et al., 2019) and others (Naninck et al., 2015; Others) have found effects of LB on depressive-like behavior in females but failed to find significant differences in anxiety-like behavior in adult mice. Ongoing work in our lab has replicated the lack of effect in auditory fear conditioning in adult mice. However, we are aware that other groups (Guadagno et al., 2018) have found significant effects of LB rearing on fear conditioning and expression. Differences between our fear conditioning results and Guadagno et al., 2018 may be due to species (differences in activity levels, learning differences, stress tolerance) or LB protocol (administration from PND 1-9 vs PND 4-11).